# WILDVISION: Evaluating Vision-Language Models in the Wild with Human Preferences

**Yujie Lu**[♠]    **Dongfu Jiang**[♡]

**Wenhu Chen**[♡]    **William Yang Wang**[♠]    **Yejin Choi**[◇♣]    **Bill Yuchen Lin**[◇]

[◇]Allen Institute of AI    [♣]University of Washington
[♠]University of California, Santa Barbara    [♡]University of Waterloo

yujielu@ucsb.edu, yuchenl@allenai.org

🤗 https://hf.co/spaces/WildVision/vision-arena

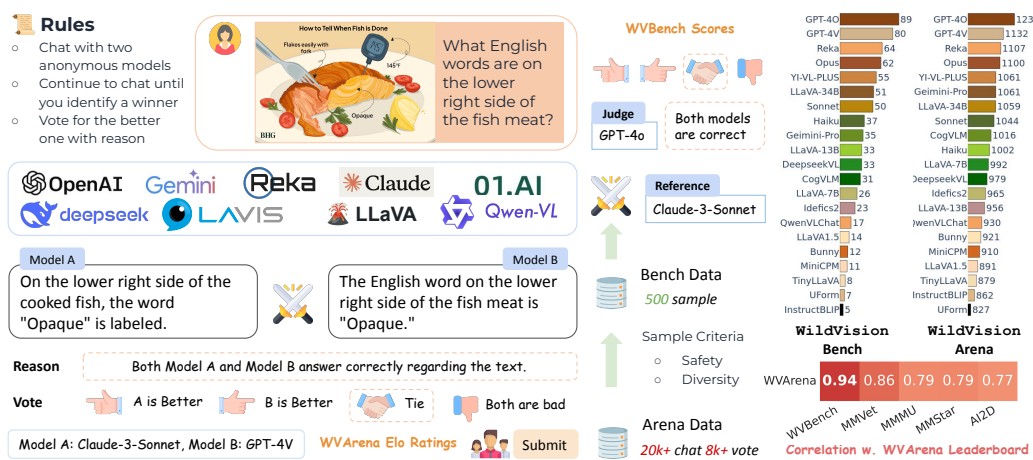

Figure 1: WILDVISION-ARENA (WV-ARENA) supports multi-round multimodal chats with $20+$ models, enabling the comparison of VLMs in real-world scenarios. We curate WILDVISION-BENCH (WV-BENCH) by selecting 500 samples from $20k+$ in-the-wild chats and $8k+$ user ratings. Automatic model scorings on WV-BENCH closely correlate with the Elo ratings on WV-ARENA.

## Abstract

Recent breakthroughs in vision-language models (VLMs) emphasize the necessity of benchmarking human preferences in real-world multimodal interactions. To address this gap, we launched WILDVISION-ARENA (WV-ARENA), an online platform that collects human preferences to evaluate VLMs. We curated WV-BENCH by selecting 500 high-quality samples from 8,000 user submissions in WV-ARENA. WV-BENCH uses GPT-4 as the judge to compare each VLM with Claude-3-Sonnet, achieving a Spearman correlation of 0.94 with the WV-ARENA Elo. This significantly outperforms other benchmarks like MMVet, MMMU, and MMStar. Our comprehensive analysis of 20K real-world interactions reveals important insights into the failure cases of top-performing VLMs. For example, we find that although GPT-4V surpasses many other models like Reka-Flash, Opus, and Yi-VL-Plus in simple visual recognition and reasoning tasks, it still faces challenges with subtle contextual cues, spatial reasoning, visual imagination, and expert domain knowledge. Additionally, current VLMs exhibit issues with hallucinations and safety when intentionally provoked. We are releasing our chat and feedback data to further advance research in the field of VLMs.

38th Conference on Neural Information Processing Systems (NeurIPS 2024) Track on Datasets and Benchmarks.

# 1 Introduction

Vision-language models (VLMs) [68, 82, 69, 49, 14, 113, 3, 5] have shown groundbreaking performance across various applications, necessitating enhanced evaluation approaches [87, 24, 107, 106] to keep up with their rapid advancements. Current evaluation benchmarks, however, are constrained by simplicity [53, 102] and practicality [101, 50]. Meanwhile, evaluation metrics for vision and language tasks are predominantly reference-based, focusing on exact matches or model-based scores [87, 7]. The success of the CLIP model [73] has enabled reference-free evaluation [24], reducing the need for reference curation while maintaining alignment with human annotators. More recent evaluation methods [56, 107, 35] leverage the instruction-following capability of LLMs and the expertise of vision models [15, 91, 34], making the automatic evaluation of VLMs more fine-grained and interpretable. Despite these advancements, a gap remains between these metrics and human preferences when comparing a large number of models' capabilities in real-world multimodal interactions.

In this paper, we introduce WILDVISION-ARENA and WILDVISION-BENCH to address the need for tracking human preferences regarding models' capabilities in the wild. Our WILDVISION-ARENA is a chatbot-style [110, 12] platform that facilitates easy comparison among VLMs, utilizing the Elo Rating system as the primary ranking metric. With the support of over 20 models (GPT-4o [69], GPT-4V [68], Gemini-Pro [82], Gemini-1.5 [81], Reka [83], Claude-3 [2], LLaVA-NEXT [48], etc), alongside a side-by-side chatting interface over images, we have crowdsourced over $20,000$ multi-round human-AI chat interactions, including over $8,000$ votes and fine-grained feedback. We then sample diversified and safe data as our WILDVISION-BENCH and adapt AlpacalEval [44] to visual context. Specifically, we use the latest released GPT-4o [69] as a judge model to vote between each VLM and the reference model Claude-3-Sonnet [2]. The statistically estimated model scores on WV-BENCH achieve a Spearman's Correlation of $0.94$ with Elo ratings in WILDVISION-ARENA.

Our comprehensive analysis of these in-the-wild chats identifies areas for improvement in recognizing visual context, spatial reasoning and imagination, and expert domain knowledge. Additionally, lower-performing VLMs struggle with discerning fine visual details in images, hindered by resolution and contextual limitations. Across the board, these models also face challenges with hallucination and safety concerns. Our main contributions can be summarized as:

- We develop WILDVISION-ARENA, an interactive evaluation platform that hosts over 20 VLMs and a live leaderboard reflecting crowdsourced user preferences on real-world chats.

- We curate WILDVISION-BENCH from WILDVISION-ARENA, a fast-evaluation benchmark that closely aligned with human preferences at $0.94$ Spearman's Correlation.

- We comprehensively analyze $20,000+$ multimodal conversations and $8,000+$ votes, and we will release this data to advance future research in VLMs.

| Statistic | Number |
|---|---|
| Total Votes | 8,076 |
| Anonymous | 6,636 |
| Non-anonymous | 1,440 |
| Left Vote | 2,932 |
| Right Vote | 2,839 |
| Tie Vote | 979 |
| Bad Vote | 1,326 |
| Days | 102 |
| Total Round | 10,884 |
| Avg Round | 1.34 |
| Avg Token Input | 31.00 |
| Avg Token Output | 108.87 |

Table 1: Statistics of votings in WV-ARENA.

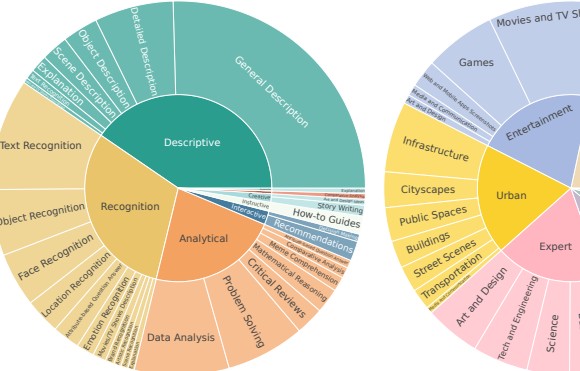

Figure 2: Question Category

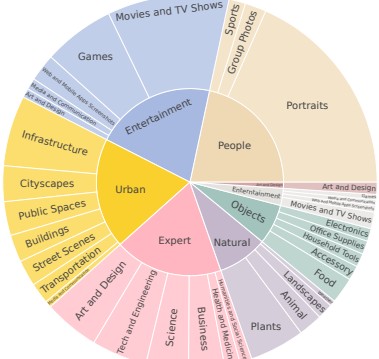

Figure 3: Image Domain

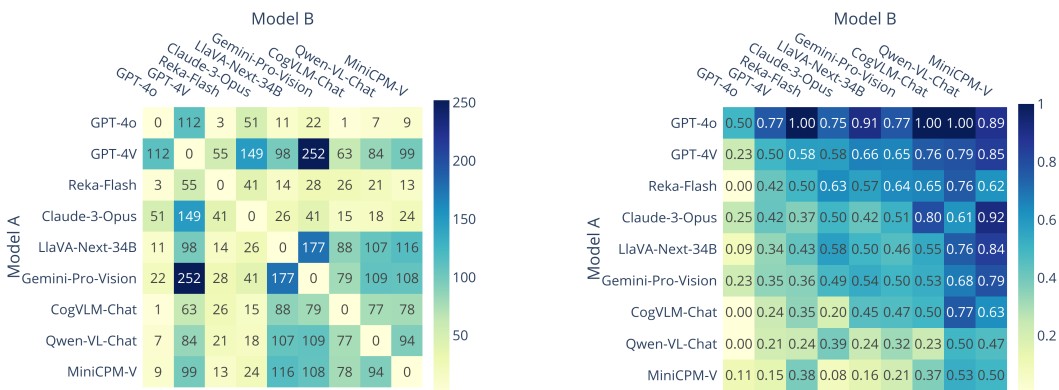

Figure 4: Battle Count Heatmap (Left): the number of voted comparisons between models. Win Fraction Heatmap (Right): the winning rate of Model A over Model B in voted comparisons.

## 2 WILDVISION-ARENA: Ranking VLMs with Human Preference

In this section, we introduce WILDVISION-ARENA and present statistics of in-the-wild chat data, along with a deep analysis of human preferences that formulate our online VLMs leaderboard.

### 2.1 Overview Design of WILDVISION-ARENA

Users conduct multi-round chats over uploaded images, during which two models from the pool or third-party APIs are sampled. Users vote for the better response, with the model's identity revealed afterward, and can provide reasons for their choices. Votes contribute to a live leaderboard, which is updated every few hours to rank the models. Appendix A shows a screenshot of our user interface. In WILDVISION-ARENA, we currently support 20+ VLMs as shown in the leaderboard on the right part of Figure 1. The generation hyperparameters are set the same when comparing these models, and users can change the temperature, top-p and max output tokens per their use cases.

### 2.2 Statistics of Chat Data with Votings

Each chat data point that has human voting is classified into a category-subcategory and domain-subdomain using GPT-4v . The prompt template details are provided in Appendix E.1. Key statistics of user voting in WILDVISION-ARENA are presented in Table 1. The number of tokens is estimated with tiktoken tokenizer corresponding to model 'gpt-3.5-turbo'. Figure 2 and Figure 3 visualize the distribution of these voting data in terms of question categories and image domains, respectively. In addition to the three dominant question categories (Recognition, Descriptive, Analytical), the Interactive, Instructive, and Creative categories are also receiving increasing interest. Users are mostly interested in chat about images tagged with the *Entertainment* domain (most of which are related to games and movies/TV shows), as well as the Urban, Expert, and People domains.

### 2.3 Crowdsourced Human Preference on VLMs in the Wild

**Pairwise Comparison**   We visualize the heatmap of battle counts and win fractions of seven models out of the 20+ models supported in the WILDVISION-ARENA in Figure 4. The battle count heatmap highlights the frequency of direct comparisons, with models like GPT-4V vs. Gemini-Pro (252 voted battles) being tested more rigorously. GPT-4o consistently outperforms the others by a large margin, winning 77% of its battles against the second-best model, GPT-4V, which ranks as the second best. Reka-Flash follows closely behind GPT-4V, winning 42% of its battles, while other models demonstrate lower winning rates. Among the open-source models, LLaVA-NEXT leads, though there remains a significant gap between it and both GPT-4V and GPT-4o.

**Expert Agreement with User Voting**   To assess the quality of crowdsourced user voting data on our platform, we evaluated inter-annotator agreement by comparing the annotations of our experts

Table 2: WILDVISION-ARENA Leaderboard. We show the full elo score and within three question categories (`Analytical`, `Descriptive`, `Recognition`) and three image domains (`Entertainment`, `Objects`, `Expert`) of 22 models with a time cutoff at May 29, 2024. **Best** Second Best Best among proprietary models Best among open-source models.

| Models | Size | Elo | Battles | MMMU | Question Category | | | Image Domain | | |
|---|---|---|---|---|---|---|---|---|---|---|
| | | | | | Analyt. | Descri. | Recogn. | Entert. | Objects | Expert |
| GPT-4O [69] | – | **1235** | 434 | **62.8** | **1290** | **1250** | **1236** | **1362** | **1203** | **1293** |
| GPT-4-Vision [68] | – | 1132 | 2288 | 56.8 | 1154 | 1169 | 1099 | 1177 | 1109 | 1178 |
| Reka-Flash [83] | – | 1107 | 513 | 56.3 | 1093 | 1141 | 1067 | 1069 | 1101 | 1191 |
| Claude-3-OPUS [2] | – | 1100 | 908 | 59.4 | 1117 | 1096 | 1092 | 1111 | 1127 | 1128 |
| Gemini-Pro-Vision [82] | – | 1061 | 2229 | 47.9 | 1099 | 1041 | 1090 | 1088 | 1077 | 1041 |
| Yi-VL-PLUS [1] | – | 1061 | 283 | – | 1084 | 1040 | 1078 | 1001 | 1119 | 1101 |
| LLaVA-NEXT [48] | 34B | 1059 | 1826 | 51.1 | 1068 | 1104 | 1021 | 1074 | 1015 | 1052 |
| Gemini-1.5-Flash [81] | – | 1055 | 132 | – | 1090 | 1018 | 1085 | 1190 | 990 | 1127 |
| Claude-3-Sonnet [2] | – | 1044 | 496 | 53.1 | 1063 | 1056 | 1041 | 1033 | 1023 | 1119 |
| CogVLM-Chat-HF [89] | 13B | 1016 | 1024 | 32.1 | 950 | 947 | 1006 | 955 | 930 | 950 |
| Claude-3-Haiku [2] | – | 1002 | 419 | 50.2 | 964 | 1008 | 996 | 1033 | 1014 | 1005 |
| LLaVA-NEXT [48] | 7B | 992 | 1367 | 35.1 | 963 | 1032 | 977 | 992 | 1023 | 1001 |
| DeepSeek-VL [51] | 7B | 979 | 646 | 36.6 | 988 | 984 | 953 | 956 | 1026 | 962 |
| Idefics2 [37] | 8B | 965 | 100 | 36.6 | 818 | 1003 | 1011 | 909 | 1071 | 1020 |
| LLaVA-NEXT [48] | 13B | 956 | 201 | 35.9 | 965 | 974 | 1006 | 975 | 971 | 987 |
| Qwen-VL-Chat [5] | 10B | 930 | 1328 | 35.9 | 898 | 937 | 940 | 923 | 942 | 902 |
| Bunny-V1 [23] | 3B | 921 | 389 | 38.2 | 897 | 922 | 878 | 884 | 823 | 823 |
| MiniCPM-V [26] | 3B | 910 | 1349 | 34.7 | 895 | 911 | 925 | 888 | 890 | 840 |
| LLaVA-v1.5 [47] | 13B | 891 | 299 | 36.4 | 952 | 838 | 920 | 887 | 827 | 914 |
| Tiny-LLaVA-v1-HF [111] | 3B | 879 | 288 | 33.1 | 901 | 828 | 821 | 808 | 853 | 894 |
| InstructBLIP [14] | 7B | 862 | 807 | 30.6 | 834 | 856 | 891 | 840 | 902 | 763 |
| UFORM-Gen2-Qwen [86] | 500M | 827 | 452 | – | 911 | 785 | 853 | 768 | 937 | 830 |

with those from users of the WILDVISION-ARENA. This analysis was conducted on a set of 100 samples. Our findings indicate a substantial level of agreement with the two experts, with an average percentage agreement of $72.5\%$. Furthermore, the calculated Cohen's Kappa coefficient was $0.59$, suggesting a moderate to high degree of reliability in the annotations across different annotators.

## 2.4 Model Ranking with Elo Rating in WILDVISION-ARENA

Following Chatbot Arena [12], we adapt Elo Rating System [17] to provide a dynamic evaluation platform for ranking VLMs by statistical modeling based on our collected direct pairwise comparisons. We briefly introduce the Online Elo Rating and the statistical estimation method.

**Online Elo Rating**   Elo rating focuses on modeling the probability of player $i$ winning against player $j$ given their existing ratings $R_i$ and $R_j$ respectively, where $i, j \in N$. We define a binary outcome $Y_{ij}$ for each comparison between player $i$ and player $j$, where $Y_{ij} = 1$ if player $i$ wins against player $j$, and $Y_{ij} = 0$ otherwise. Then the logistic probability is formulated as:

$$P(Y_{ij} = 1) = \frac{1}{1 + 10^{(R_j - R_i)/\alpha}}, \tag{1}$$

where $\alpha = 400$ for Elo rating computation. After a match, each player's rating is updated by the formula: $R'_i = R_i + K \times (S(i|j) - E(i|j))$, where $S(i|j)$ is the actual match outcome (1 for a win, 0.5 for a tie, and 0 for a loss), and $E(i|j) = P(Y_{ij} = 1)$. The higher-rated player will win fewer points if they win but lose more if they lose, while the lower-rated player will experience the opposite. The computation of the online Elo rating is correlated with the comparison order. Therefore, we follow Chatbot Arena to adopt the Bradley–Terry model [9] for a stable statistical estimation.

The probability of player $i$ winning against player $j$ given their existing ratings $R_i$ and $R_j$ respectively.

**Statistical Estimation**   The Bradley–Terry model [9] estimates the Elo rating using a logistic regression model and maximum likelihood estimation (MLE). Let's say there are $N$ players, and we have a series of pairwise comparisons, where $W_{ij}$ is the number of times player $i$ wins against player $j$. The log-likelihood function for all pairwise comparisons can be written as:

$$\mathcal{L}(\mathbf{R}) = \sum_{i,j \in N, i \neq j} \left( W_{ij} Y_{ij} \log P(Y_{ij} = 1) \right), \tag{2}$$

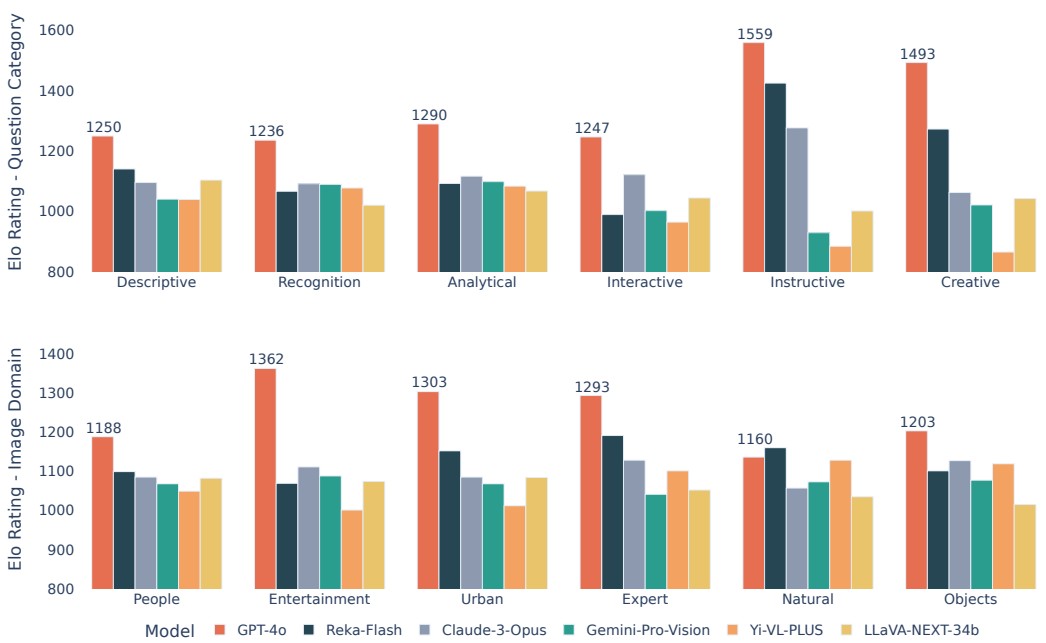

Figure 5: Elo ratings of six models across question categories (Top) and image domains (Bottom).

where $\mathbf{R} = \{R1, ..., R_N\}$ is the Elo rating variable of each player. Since this modeling does not consider ties, in practice, we duplicate all the votes and force half of the tie votes to be counted as left model $i$ winning ($Y_{ij} = 1$) and the other half as right model $j$ winning ($Y_{ij} = 0$).

## 2.5 WILDVISION-ARENA Leaderboard

We report the leaderboard results in Table 2, including the full Elo ratings and the total number of battles for each model, with a time cutoff on May 29, 2024. Additionally, we provide the Elo ratings for three main question categories (`Analytical`, `Descriptive`, `Recognition`) and three main image domains (`Entertainment`, `Natural`, `Expert`) to better understand the specialties of each model. GPT-4o quickly dominates the leaderboard after its release, surpassing the previous state-of-the-art GPT-4V by a significant margin, followed by Reka-Flash, Claude-3-OPUS. Yi-VL-PLUS and LLaVA-NEXT-34B achieve the same rank, reflecting that both models are based on the Yi [1]. Among open-source models, LLaVA-NEXT-34B ranks first, even surpassing Gemini-1.5-Flash and Claude-3-Sonnet, Claude-3-Haiku, indicating a strong baseline for research purposes. To compare models under each question category and image domain, we present the top six models ranked in the WILDVISION-ARENA leaderboard in terms of Elo ratings for each question category and image domain in Figure 5. GPT-4o consistently outperforms all other models except for the images tagged with `Natural`, where varying specialties are more commonly observed among the other models.

## 3 WILDVISION-BENCH: In-the-Wild Testbed for VLMs

Recent VLMs reveal a closing gap with GPT-4V on various benchmarks[101, 102], but this improvement is not always reflected in users' daily experiences. This discrepancy arises from current models' limited generalizability compared to proprietary ones, which fixed benchmarks fail to capture. To address this, we propose creating WILDVISION-BENCH, a challenging and natural benchmark for VLMs that reflects real-world human use cases, with models' rankings aligning closely with the WILDVISION-ARENA leaderboard contributed by diverse crowdsourced user votes.

Table 3: VLMs' responses on two cases from WILDVISION-BENCH expert annotated samples. The example #61 is a hard case that all models fall short at.

*Image* #44 in WildVision Bench

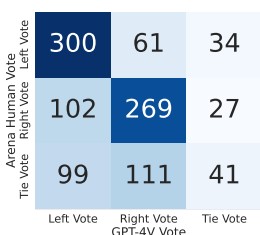

**Text Prompt:** Answer shortly what is he doing? And what do we call him?

**GPT-4V:** He is performing a breakdance move. We call him a breakdancer or b-boy.
**LLaVA-v1.6-34B:** He is performing a dance move that involves spinning or flipping while balancing on one hand ..., he may be referred to as a dancer or acrobat.
**Gemini Pro Vision:** He is breakdancing. He is called a b-boy or breaker.

*Image* #61 in WildVision Bench

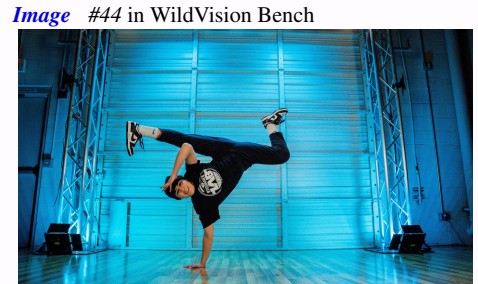

**Text Prompt:** Convert the first row of this sudoku matrix into the text format, each cell separated by a space. Use 0 to represent empty cells.

**GPT-4V:** 6 8 4 2 1 0 0 0 0

**LLaVA-v1.6-34B:** 6 8 4 0 2 1 0 7 3 4 0 6

**Gemini Pro Vision:** 6 8 4 2 1 9 8 7 7

## 3.1 Data Curation Pipeline

Starting with in-the-wild multimodal conversation data from WILDVISION-ARENA's users, we apply the NSFW detector [36] on the images to filter out unsafe content. We then perform deduplication on the images and apply diversity sampling to formulate a public set of 500 data samples for WILDVISION-BENCH. Our experts manually annotate 50 samples as a preview of a hidden set, which will be updated dynamically to avoid contamination. We showcase the model performance on two cases from expert annotations in Table 3.

## 3.2 Automatic Evaluation on WILDVISION-BENCH

| Metric vs Human | GPT-4v | | |
|---|---|---|---|
| | **4-way** | **3-way** | **Binary** |
| F1 Score (Macro) | 0.4245 | 0.5143 | 0.7792 |
| F1 Score (Micro) | 0.5747 | 0.5842 | 0.7796 |
| F1 Score (Weighted) | 0.5407 | 0.5536 | 0.7798 |
| Cohen's Kappa Score | 0.3404 | 0.3442 | 0.5585 |
| Pearson Correlation | 0.2906 | 0.2880 | 0.5587 |

Figure 6: Left: GPT-4V vs. Arena Human Voting. Right: Agreement; 4-way: left/right/tie/bad vote. 3-way: left/right/other. Binary: left/right vote

**VLMs as a Local Evaluator** Previous work [107, 35] shows alignment between GPT-4V and humans when evaluating the performance of VLMs. We further validate the agreement of GPT-4V with crowdsourced human preferences in WILDVISION-ARENA to ensure its efficacy in the wild. Specifically, we feed a pair of multimodal conversations along with the votes into GPT-4V to select among four choices: 1) left/right vote: the left/right model response is better, 2) tie/bad vote: both models are equally good/bad. In Appendix E.3, we provide the detailed prompt template for GPT-4V. We show the GPT-4V vs Arena Human alignment in Figure 6. We observe that GPT-4V has relatively low agreement with humans on tie votes but shows high agreement with humans when both models

Table 4: Estimated model scores of VLMs on WILDVISION-BENCH test split of 500 samples.

| Model | Score | 95% CI | Win Rate | Reward | Much Better | Better | Tie | Worse | Much Worse | Avg Tokens |
|---|---|---|---|---|---|---|---|---|---|---|
| GPT-4o [69] | 89.41 | (−1.7, 2.0) | 80.6% | 56.4 | 255.0 | 148.0 | 14.0 | 72.0 | 11.0 | 157 |
| GPT-4-Vision [68] | 80.01 | (−1.9, 2.8) | 71.8% | 39.4 | 182.0 | 177.0 | 22.0 | 91.0 | 28.0 | 140 |
| Reka-Flash [83] | 64.79 | (−2.9, 3.0) | 58.8% | 18.9 | 135.0 | 159.0 | 28.0 | 116.0 | 62.0 | 181 |
| Claude-3-Opus [2] | 62.15 | (−2.8, 3.4) | 53.0% | 13.5 | 103.0 | 162.0 | 48.0 | 141.0 | 46.0 | 120 |
| Yi-VL-PLUS [1] | 55.09 | (−2.9, 3.0) | 52.8% | 7.2 | 98.0 | 166.0 | 29.0 | 124.0 | 83.0 | 150 |
| LLaVA-NEXT-34B [48] | 51.91 | (−3.1, 2.4) | 49.2% | 2.5 | 90.0 | 156.0 | 26.0 | 145.0 | 83.0 | 165 |
| Claude-3-Sonnet [2] | 50.00 | − | − | − | − | − | − | − | − | 120 |
| Claude-3-Haiku [2] | 37.70 | (−3.2, 4.2) | 30.6% | −16.5 | 54.0 | 99.0 | 47.0 | 228.0 | 72.0 | 97 |
| Gemini-Pro-Vision [82] | 35.45 | (−2.6, 3.2) | 32.6% | −21.0 | 80.0 | 83.0 | 27.0 | 167.0 | 143.0 | 66 |
| LLaVA-NEXT-13B [48] | 33.69 | (−3.8, 2.7) | 33.8% | −21.4 | 62.0 | 107.0 | 25.0 | 167.0 | 139.0 | 138 |
| DeepSeek-VL-7B [51] | 33.48 | (−2.2, 3.0) | 35.6% | −21.2 | 59.0 | 119.0 | 17.0 | 161.0 | 144.0 | 119 |
| CogVLM-Chat-HF [89] | 31.88 | (−2.7, 2.4) | 30.6% | −26.4 | 75.0 | 78.0 | 15.0 | 172.0 | 160.0 | 63 |
| LLaVA-NEXT-7B [48] | 26.15 | (−2.7, 2.3) | 27.0% | −31.4 | 45.0 | 90.0 | 36.0 | 164.0 | 165.0 | 139 |
| Idefics2 [37] | 23.71 | (−2.4, 2.5) | 26.4% | −35.8 | 44.0 | 88.0 | 19.0 | 164.0 | 185.0 | 128 |
| Qwen-VL-Chat [5] | 17.87 | (−2.6, 2.2) | 19.6% | −47.9 | 42.0 | 56.0 | 15.0 | 155.0 | 232.0 | 70 |
| LLaVA-v1.5-13B [47] | 14.15 | (−2.2, 2.2) | 16.8% | −52.5 | 28.0 | 56.0 | 19.0 | 157.0 | 240.0 | 87 |
| Bunny-3B [23] | 12.70 | (−1.8, 1.9) | 16.6% | −54.4 | 23.0 | 60.0 | 10.0 | 164.0 | 243.0 | 76 |
| MiniCPM-V [26] | 11.66 | (−1.8, 2.1) | 13.6% | −57.5 | 25.0 | 43.0 | 16.0 | 164.0 | 252.0 | 89 |
| Tiny-LLaVA [111] | 8.01 | (−1.4, 1.4) | 11.0% | −66.2 | 16.0 | 39.0 | 15.0 | 127.0 | 303.0 | 74 |
| UFORM-Gen2-Qwen [86] | 7.55 | (−1.6, 1.1) | 10.8% | −68.5 | 16.0 | 38.0 | 11.0 | 115.0 | 320.0 | 92 |
| InstructBLIP-7B [14] | 5.54 | (−1.3, 1.5) | 7.8% | −72.5 | 11.0 | 28.0 | 15.0 | 117.0 | 329.0 | 47 |

exhibit distinguishable differences. However, predicting when both models are bad is challenging as GPT-4V sometimes falls short in these examples as well.

**WILDVISION-BENCH Alignment with Human Preferences in WILDVISION-ARENA** Inspired by Alpaca Eval [16], we adopt a similar approach to rank VLMs on our WILDVISION-BENCH automatically. Specifically, we use GPT-4o as the judgment model and Claude-3-Sonnet as our reference model. We compare each model's answers on the WILDVISION-BENCH public set with Claude-3-Sonnet and then use GPT-4o, which shows better alignment with humans in our cases, to give a vote. The template in Table E.3 is used for the prompt of the judge, where 5 levels of comparison results are defined, which are "Better+", "Better", "Tie", "Worse", and "Worse+" respectively. We report the score results of these models in Table 4. This achieves a 0.94 Spearman correlation with the WILDVISION-ARENA leaderboard.

**Benchmark Correlation Heatmap** We visualize the Spearman correlation heatmap among various multimodal benchmarks in Figure 7. The MMBench-series [50] (CCBench, MM-Bench EN, MMBench CN) considers fine-grained perception and reasoning tasks in multiple choice questions. MMVet [101] evaluates integrated capabilities in visual question answering. MMStar [10] alleviates misjudgment issues with high-quality multiple choice questions. HallucionBench [22] focus on investigating hallucination issues, while MMMU [102] and MathVista [53] focus on college-level subject knowledge and mathematical reasoning in visual contexts, respectively. WildVision Elo represents the arena leaderboard, reflecting human preferences using Elo ratings from pairwise comparisons. WildVision Bench represents ranking model using estimated model score on our WILDVISION-BENCH. This achieves the highest correlation with WildVision Elo, indicat-

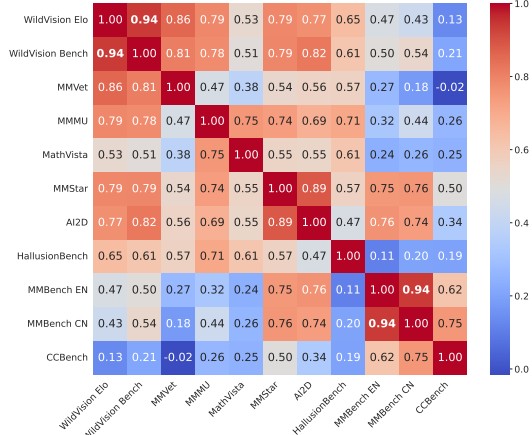

Figure 7: WILDVISION-BENCH achieves the highest correlation with WILDVISION-ARENA, with a Spearman's correlation of 0.94.

ing its crucial role in simulating human preferences on these VLMs in the real world. The runner-up in alignment with human preferences is MMVet, followed by MMMU and MMStar.

Table 5: Failure cases of GPT-4V and Gemini-Pro-Vision sampled from WILDVISION-ARENA.

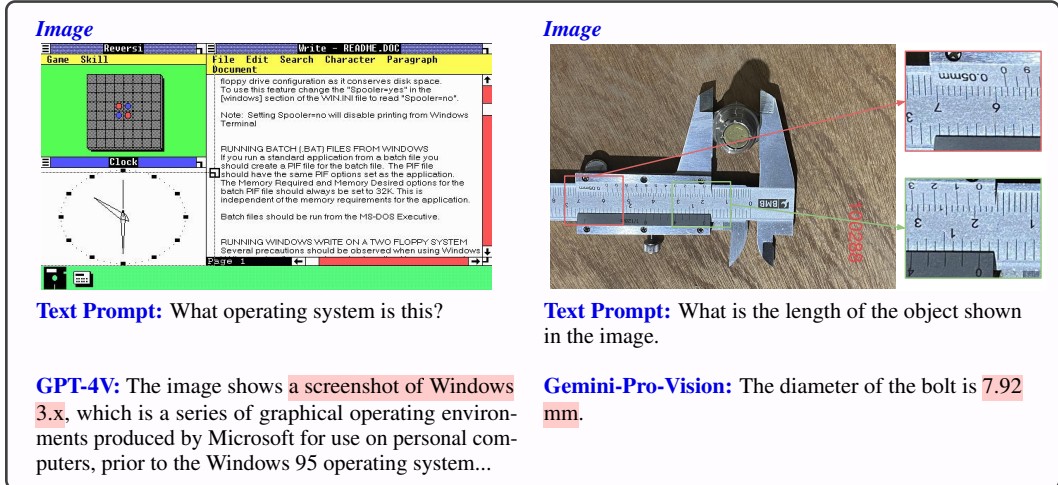

**Text Prompt:** What operating system is this?

**GPT-4V:** The image shows a screenshot of Windows 3.x, which is a series of graphical operating environments produced by Microsoft for use on personal computers, prior to the Windows 95 operating system...

**Text Prompt:** What is the length of the object shown in the image.

**Gemini-Pro-Vision:** The diameter of the bolt is 7.92 mm.

## 4 Analysis

**In-the-wild Multimodal Chat**    In contrast to public benchmark, in-the-wild multimodal conversations involve images and instructions from a diverse range of sources and receive vote data from a varied group of users. This better helps us understand how current VLMs can benefit real-world scenarios and reveal improvement directions for researchers in the field. In Appendix B, we present more cases under each image domain and question category. We will release both multimodal chat and crowdsourced voting data for future research.

**Failure Cases**    In Table 5, we present two distinct failure instances that are documented in the WILDVISION-ARENA platform. This analysis reveals that GPT-4V's limitations primarily stem from insufficient background knowledge, whereas Gemini-Pro-Vision often fails to discern and process subtle details crucial for deriving correct answers. Additional details on these failure cases are provided in Appendix Our categorization of common failures includes six types: Visual Recognition, Visual Reasoning, Spatial Imagination, Contextual Understanding, Expert Domain Knowledge, Hallucination, and Safety. Although not all failure cases can be included in this paper, we plan to periodically release additional cases on our live platform to aid ongoing research and development.

**Model Comparison on WILDVISION-BENCH**    Table 3 compares the responses of GPT-4V, LLaVA-NEXT-34B, and Gemini-Pro-Vision on a validation sample from WILDVISION-BENCH. GPT-4V generally outperforms the other models, confirming expectations of its superior capabilities. Nevertheless, all models occasionally fail to deliver correct responses, notably in scenarios requiring compositional reasoning, regardless of the simplicity of the text or the image involved. We also observe that recognizing and interpreting subtle visual details within images is still challenging for less capable models.

**Broader Impact**    For the first version of data release, we plan to release over 20,000 crowdsourced multi-turn conversation data and more than 8,000 human votings with reasons, providing a valuable resource for understanding human preferences in VLMs interactions and developing models that align more closely with human standards in real-world scenarios. We will also present a live leaderboard together with useful failure case analysis to keep track of recent advancements in this field. Additionally, by open-sourcing the WILDVISION-ARENA code, we enable researchers and developers to adapt our methods to other domains. We will also support fast evaluation of our WILDVISION-BENCH for quick and human-aligned evaluation, which aligns with the human preferences in VLMs in real-world scenarios.

**Modality, Resolution, Long Context, Resource-Efficent**   Many work have extended vision-language models (VLMs) beyond image-text modalities, including video [105, 57, 109], audio [13], and even applied to embodied agent [65]. Future work may consider improving all-in-one models [63, 92, 82, 112, 19] by discovering better methods to integrate these modality data. Recent works have enabled high-resolution [48, 96] and text reading [108, 25] capabilities in VLMs, although many failure cases are still induced by low resolution or poor OCR capability. Other work advances multi-image and long-context capabilities in VLMs [61, 37, 29, 79, 54]. We expect future research to discover the best mechanisms for balancing compact and effective approaches to convey multimodal information, such as recent progress of text representation in pixel space [75, 18, 55]. This is essential to closing the gap between open-source multimodal agents [99, 104] and proprietary ones [97, 69]. Although many works [26, 111] have made VLMs more compact, their performance is still not satisfying. Future work may further improve the performance of smaller models with less training data and higher throughput inference.

**World Knowledge and Safety in VLMs**   The challenge of embedding extensive world knowledge within VLMs is significant, particularly given their current limitations in understanding physical principles and interacting with real-world environments. These models' ability to dynamically expand their knowledge base through activities like browsing the internet, reading books, or watching videos is an exciting potential advancement. Key concerns in LLMs include security [94, 64, 90, 98], privacy [31, 38], and the propagation of truthfulness [30, 77, 45] and prevention of misinformation [80, 72, 103]. For VLMs, they face unique safety challenges: 1) incorrect alignment of multimodal data can lead to harmful outputs, 2) images may contain sensitive information, necessitating careful handling, and 3) VLMs are vulnerable to attacks manipulating both text and images.

# 5   Related Work

**Live Benchmarking for vision-language models**   Vision-and-language pre-training starts from models [42, 43] adapting objectives in BERT [33], to models [74] adopting contrastive learning, and to unified frameworks [52, 88, 41, 40] without task-specific head. With recent advancements of Large Language Models [67, 20, 4, 84, 85], their multi-modal counterparts [68, 82, 14, 113, 49, 47, 5, 28, 37] are dominating vision and language tasks. Beyond previous task-specific caption [11, 78], visual question answer [62, 59, 27, 21, 60], grounding [46, 100, 66, 58, 71], more benchmarks [101, 50, 39, 32] are proposed to capture VLMs capabilities. When building such benchmarks, there is an urge need to consider alleviating data contamination [76, 6] during eval, assuring robustness [55] and difficulty [70], and incorporating real-world scenarios [8, 93]. We build WILDVISION-ARENA to support diversified, difficult, in-the-wild, live benchmarking [12, 95] of VLMs.

**Human-Aligned Evaluation for vision-language models**   Evaluation for open-ended vision and language tasks [8, 93, 70] are usually challenging, and recent techniques improve human alignment by mapping free-form predictions to pre-defined choices [50], using larger models as the evaluator [56, 107]. In the domain of evaluating LLMs, a certain approaches [110, 16] prove their effectiveness in aligning with real-world annotators on the Chatbot Arena [12]. This inspires our efforts in curating in-the-wild small-scale WILDVISION-BENCH, that can support fast evaluation by pair-wise comparison with reference model (such as Claude-3-Sonnet [2]), and achieve alignment with crowdsourced human rators on WILDVISION-ARENA.

# 6   Conclusion

We first introduce WILDVISION-ARENA, a dynamic evaluation platform for comparing vision-language models (VLMs) in the wild. We conduct comparative insights across over 20 models by utilizing an extensive dataset of 20,000+ multimodal conversations and 8,000+ votes, allowing for continuous refinement of VLMs performance. From these in-the-wild chats, we then sample safe and diversified data for WILDVISION-BENCH and apply automatic evaluation that closely aligns with crowdsourced human preferences from WILDVISION-ARENA. Our comprehensive analysis on these in-the-wild chats indicates future directions for advancing VLMs.

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

# Part I

# Appendix

## Table of Contents

# A    User Interface

In Figure 8, we show a screenshot of the user interface of our WILDVISION-ARENA, which presents an interactive environment for evaluating multimodal large language models. This environment allows users to input questions and compare responses from multiple models simultaneously. Each model's answer is displayed side-by-side, enabling a straightforward comparison of their performance and capabilities based on user queries related to specific images or tasks. The interface also facilitates easy selection and voting to decide which model's response fits the user's criteria best, enhancing the user's ability to judge and refine the models' outputs effectively.

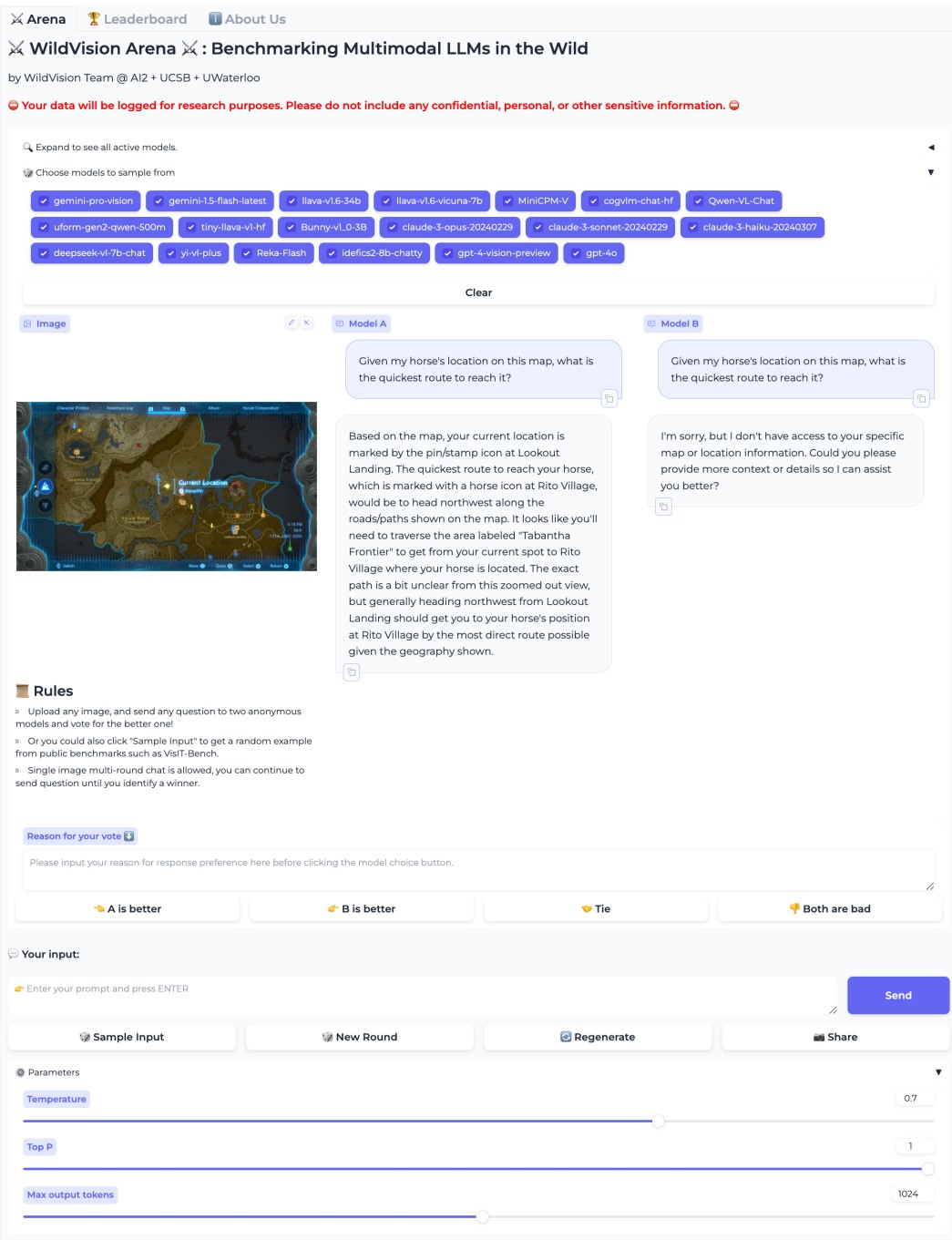

Figure 8: User Interface of WILDVISION-ARENA.

# B Question Category and Image Domain

In Table 6- 8, we showcase example data under each of the image domain and question category from WILDVISION-ARENA's users.

Table 6: Example input data in WILDVISION-ARENA tagged with [Image Domain-Subdomain] and [ Question Category-Subcategory].

*Image* [Entertainment-Movies/TV Shows]

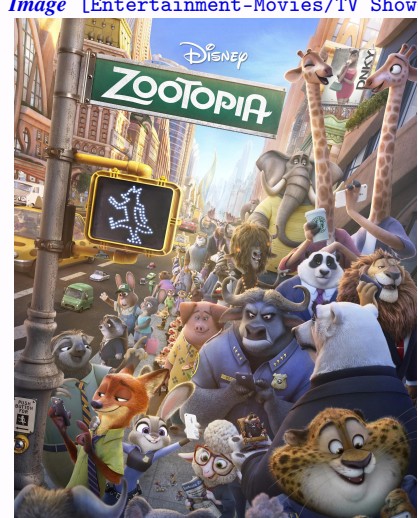

*Image* [Natural-Plants]

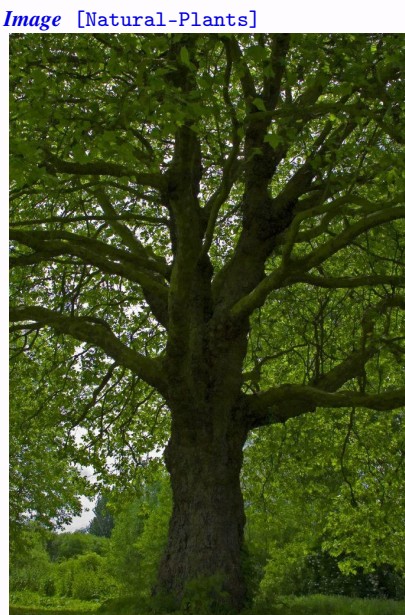

[Descriptive-Movies/TV Shows] **Text Prompt:** What are the two giraffe characters on this movie poster doing?

[Analytical-Problem Solving] **Text Prompt:** How likely is it to snow after this picture was taken? What would change with this type of tree before it's likely to snow?

*Image* [Expert-Business]

[Analytical-Data Analysis] **Text Prompt:** Which of the companies featured in the dashboard are headquartered outside the US?

*Image* [Urban-Infrastructure]

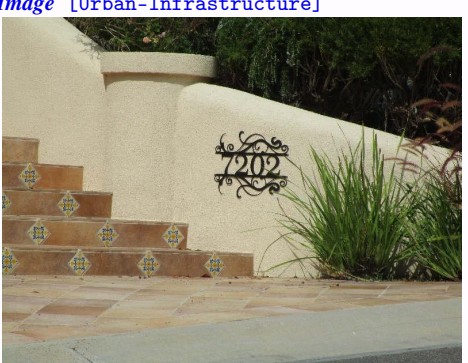

[Recognition-Text] **Text Prompt:** Can you tell me the potential risks and the unreasonale parts in the image?

Table 7: Example input data in WILDVISION-ARENA tagged with [Image Domain-Subdomain] and [ Question Category-Subcategory].

*Image* [Entertainment-Comics]

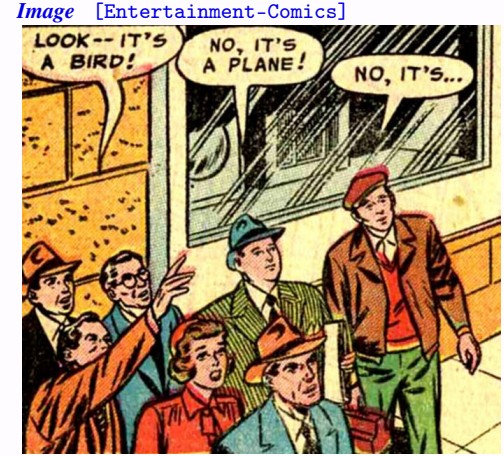

[Descriptive-Scene Description] **Text Prompt:** Whos's in the sky?

*Image* [People-Portraits]

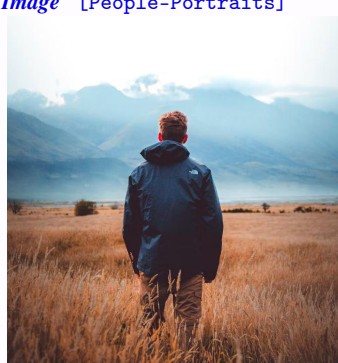

[Creative-Media Post] **Text Prompt:** write a social media post with the provided image, saying that I am ready for the new challange.

*Image* [Urban-Buildings]

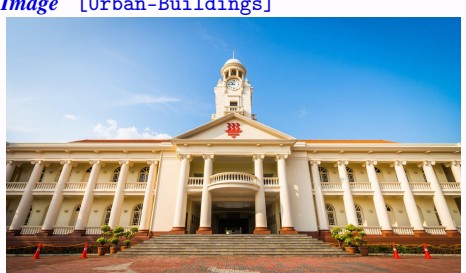

[Recognition-Location] **Text Prompt:** where is this?

*Image* [Expert-Science]

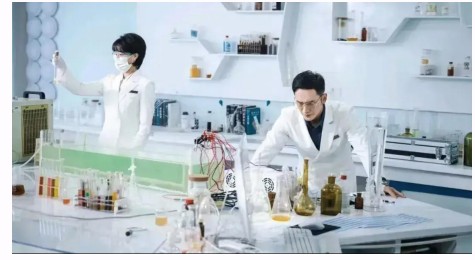

[Analytical-Safety Procedures] **Text Prompt:** Can you tell me the potential risks and the unreasonale parts in the image?

*Image* [Natural-Landscapes]

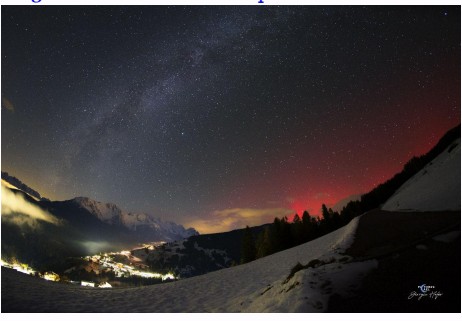

[Recognition-Location] **Text Prompt:** where was this photo taken?

*Image* [Objects-Household Tools]

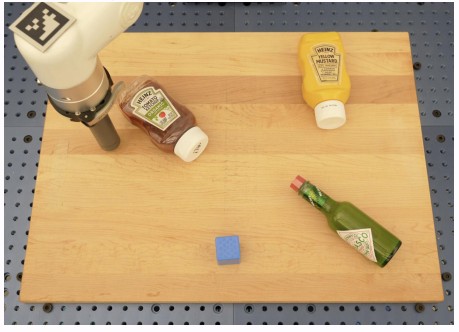

[Descriptive-Object Description] **Text Prompt:** describe the scene and objects

Table 8: Example input data in WILDVISION-ARENA tagged with [Image Domain-Subdomain] and [ Question Category-Subcategory].

***Image*** [Entertainment-Web and Mobile Apps Screenshots]

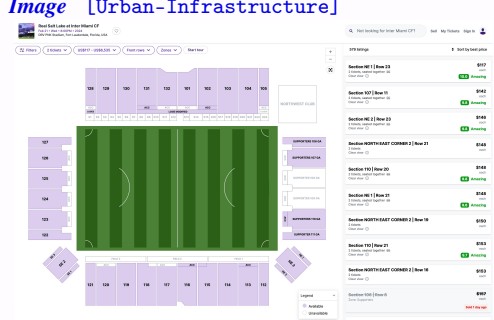

[Interactive-Web Navigation] **Text Prompt:** I need to download flyer, you will be given screenshot from browser with elements marked with number. give next action to take on web page to download the flyersngive me response in below format example 1 action:[click,scroll,wait], box:1 format action:, box:

***Image*** [Event-Sports]

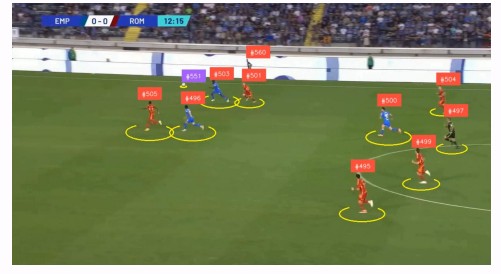

[Descriptive-Scene Description] **Text Prompt:** this is a football match , every player has an identifier , describe every player action (example : player #501 is running)

***Image*** [Urban-Infrastructure]

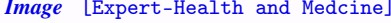

[Interactive-Recommendations] **Text Prompt:** Which section's ticket would you recommend I purchase?

***Image*** [Expert-Science]

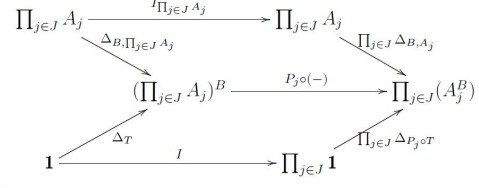

[Interactive-Code Generation] **Text Prompt:** Give me Latex code to create this diagram

***Image*** [Expert-Health and Medcine]

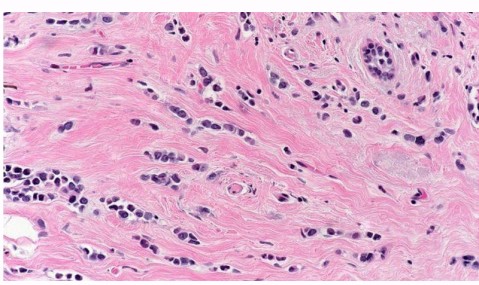

[Recognition-Object] **Text Prompt:** what type of tumor is this?

***Image*** [Entertainment-Web and Mobile Apps Screenshots]

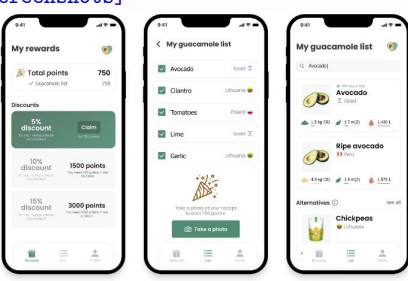

[Analytical-Critical Reviews] **Text Prompt:** Review each screenshot carefully, focusing on different aspects of usability...

# C  Analysis of Failure Cases

We observe some common failure patterns of VLMs in the wild from WILDVISION-ARENA chat data. In Tables 9- 13, we present specific failure cases. Based on the types of errors, we have condensed six categories, detailed in the following paragraphs.

**Visual Recognition**    Failures in this category involve several types of recognition challenges. Complex text, such as artistic Chinese characters (Error *#5* in Table 10), and small details, such as the text on a shop sign in a restaurant scene (Error *#3* in Table 9), often elude accurate detection. Errors also occur with small text in screenshots (Error *#18* in Table 13), recognizing objects against novel backgrounds (Error *#9* in Table 11), and identifying new objects within known contexts, like a helmet next to a motorcycle (Error *#11* in Table 11). Failures in interpreting charts (Error *#4* in Table 9) and identifying reflections (Error *#20* in Table 13) are also noted.

**Visual Reasoning**    This category covers the ability to discern visual differences and reason through visual data. Errors include overlooking changes in clothes of the character (Error *#6* in Table 10), strategizing in chess (Error *#1* in Table 9), and reading analog clocks (Error *#16* in Table 11).

**Spatial Imagination**    Challenges in this area involve imagining the outcomes of spatial transformations. Failures are evident in cube folding tasks (Error *#10* in Table 11) and visual riddles involving the folding of objects like magazines (Error *#12* in Table 11).

**Contextual Understanding**    Errors in this category relate to recognizing and understanding contexts specific to certain domains, such as movies (Error *#7* in Table 10) and video games (Error *#8* in Table 10). Failures also include responding inappropriately to marked images (Error *#17* in Table 13).

**Expert Domain Knowledge**    This involves specialized knowledge areas where VLMs fail to provide accurate information or analysis, such as verifying architectural compliance (Error *#2* in Table 9) and analyzing programming code details (Error *#15* in Table 11).

**Hallucination**    Errors of hallucination occur when models generate incorrect or misleading content based on subtle cues or nonexistent details, such as fictitious characters (Error *#13* in Table 12) or obscure brand names (Error *#18* in Table 13).

**Safety**    Issues in this category are critical as they involve handling harmful, biased, or inappropriate content, and dealing with NSFW images in ways that either excessively censor or insufficiently filter content. These images are not presented and are filtered out in benchmark curation.

Table 9: Failure Cases.

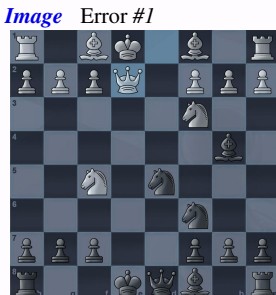

***Image***  Error *#1*

**Text Prompt:** What should I play in this position?

**GPT-4V:** This chess position is unconventional and humorous because it features a board filled with knights (horses) except for a single queen. In a standard game of chess, this situation would not occur, as each player starts with just two knights. In this specific position, there's not much strategy to discuss because the game doesn't reflect an actual playable scenario. However, if we want to engage with this as a fun exercise in imagination, you could consider moving one of the knights that isn't pinned or creating a threat to the queen. For example, you could move one of the knights on the third rank to attack the queen, such as Nc3-d5 or Ne3-c4. Remember, this is just a playful scenario and not representative of a legitimate chess position. In a real game, the best move would depend on a much broader context of the board and the strategic positions of all the pieces, not just an army of knights.

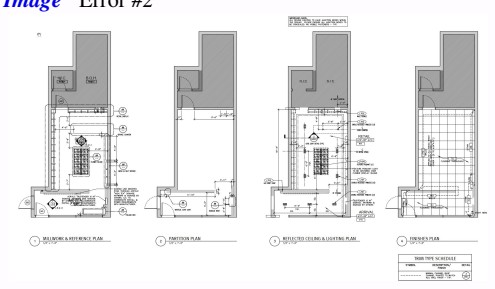

***Image***  Error *#2*

**Text Prompt:** Based on the architecture plans is this compliant in the Ontario building code

**GPT-4V:** I'm sorry, but as an AI, I do not have the capacity to verify the compliance of architectural plans with specific building codes. Building code compliance typically involves a wide range of considerations, including but not limited to accessibility, fire safety, structural integrity, and mechanical systems. Professional architects and engineers, often alongside local code enforcement officials, ensure that designs and construction meet the appropriate codes and regulations. To determine compliance with the Ontario Building Code, the plans would need to be reviewed by a certified professional who has access to and understands the local regulations and requirements. They would check if the design meets all necessary stipulations for things such as room sizes, egress requirements, bathroom accessibility, fire ratings, and many other details. You should consult with a licensed Ontario architect or a building code consultant for an official assessment of compliance.

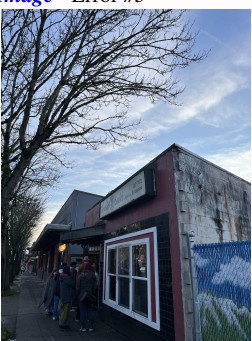

***Image***  Error *#3*

**Text Prompt:** What food will these people have very soon?

**LLaVA-NEXT-34B:** Based on the image, the people are lined up outside a building with a sign that reads "Salsa's Authentic Mexican Restaurant." It seems likely that they will be having Mexican food very soon, which could include items such as tacos, burritos, enchiladas, chiles rellenos, guacamole, and other similar dishes that are typically served at Mexican restaurants. The specific menu items will depend on the restaurant's offerings and the preferences of the customers.

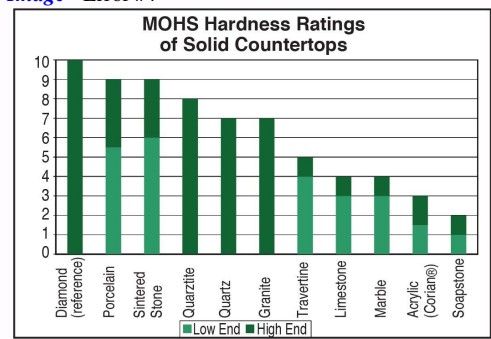

***Image***  Error *#4*

**Text Prompt:** Based on this chart, what is the MOHS hardness of Marble?

**LLaVA-NEXT-34B:** The MOHS hardness of Marble, as indicated on the chart, is 3.

Table 10: Failure Cases.

*Image* Error #5

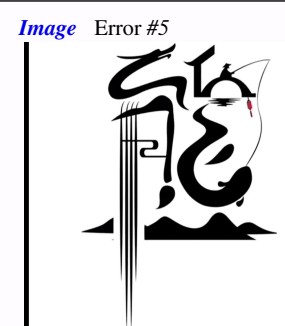

**Text Prompt:** 图片中是什么字

**GPT-4V:** 图片中的字体设计巧妙地融合了艺术与文字，呈现出既像文字又像艺术作品的效果。这个特殊的设计使得文字的具体含义不太直观，没有提供清晰的线索来解释这些字符的标准解读。如果你有更多关于这些字的背景信息或上下文，我可以试图进行更多的解释。不过，从图像中直接解读标准文本是非常困难的。

*Image* Error #6

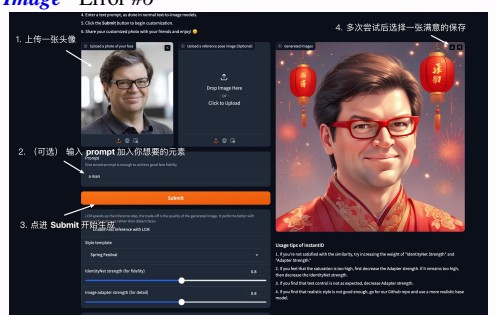

**Text Prompt:** 这张图里右侧的照片和左侧相比有什么变化

**Gemini-Pro-Vision:** 右侧的照片中，人物的脸部表情更加丰富，并且背景中添加了春节的元素，使画面更具喜庆的氛围。

*Image* Error #7

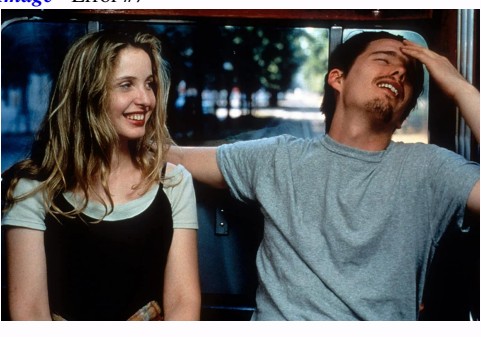

**Text Prompt:** which movie is this picture from? what is it about?

**LLaVA-NEXT-34B:** This image appears to be from the movie "The Girl"(2012), which is a psychological thriller film directed by David Riker and starring Abbie Cornish and Caleb Landry Jones. The film is about a young woman who is hired to care for a girl living in a remote Texas farmhouse, but soon discovers that the girl is not what she seems. The story revolves around themes of isolation, identity, and the consequences of one's actions.

*Image* Error #8

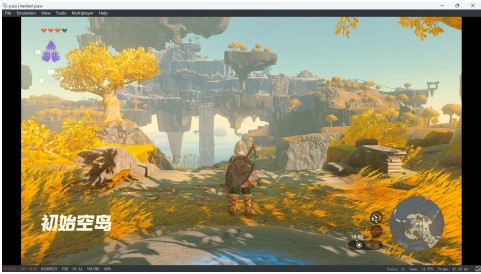

**Text Prompt:** do u know which game i am playing?

**LLaVA-NEXT-34B:** You are playing "The Legend of Zelda: Breath of the Wild,"which is an action-adventure game developed and published by Nintendo for the Wii U and Nintendo Switch consoles. The game is known for its open-world exploration and its unique gameplay mechanics.

Table 11: Failure Cases.

**Image** *Error #9*

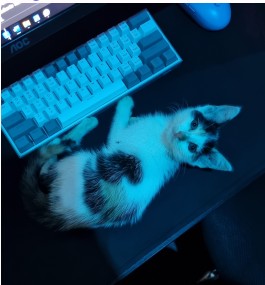

**Text Prompt:** How many keys does the keyboard have?

**GPT-4V:** The keyboard in the image features a non-standard layout, which appears to be a compact form factor lacking the numeric keypad commonly found on full-sized variants. Without being able to view the entire keyboard, it's impossible to provide an exact count of the keys...

**Image** *Error #10*

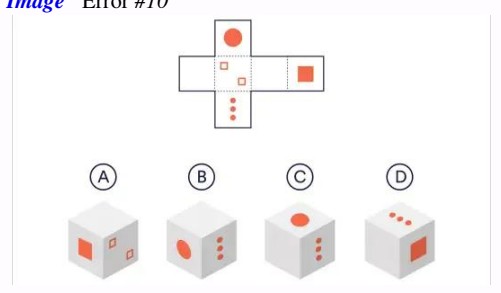

**Text Prompt:** By examining the unfolded cube (net) at the top of the image, which one of the four cubes labeled A, B, C, and D can be created by folding this net along the edges?

**Gemini-Pro-Vision:** The correct answer is C. The unfolded cube (net) at the top of the image has two red squares and three orange circles. Cube C has two red squares and three orange circles in the same arrangement as the unfolded cube.

**Image** *Error #11*

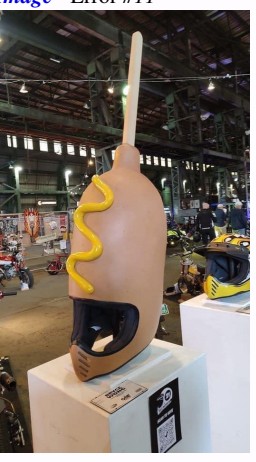

**Text Prompt:** what is this

**GPT-4V:** This is an oversized, artistic representation of a classic hand-held manual egg beater or whisk. It's designed to mimic the appearance of a traditional egg beater with an exaggerated size, likely for decorative or exhibit purposes as suggested by the setting that appears to be a show or exhibition hall. The inclusion of this piece in a public space may indicate it being part of an art installation, event display, or perhaps a promotion related to culinary equipment or innovation.

**Image** *Error #12*

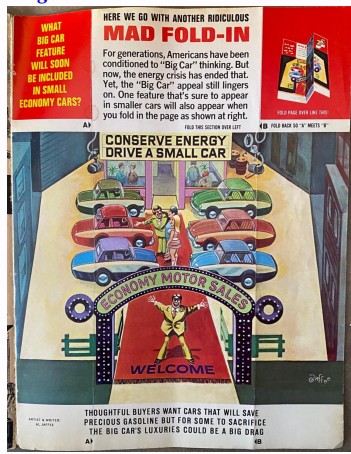

**Text Prompt:** Try to guess the solution to this Fold-in from Mad magazine. A Mad Fold-in is a visual riddle that a human can easily solve by physically folding the page as explained in the instructions in the top-right corner. I know the solution. I want to verify if you are able to solve it. What big car feature will soon be included in small economy cars?

**LLaVA-NEXT-34B:** The Mad Fold-in features a car salesman in front of a sign that says "CONSERVE ENERGY DRIVE A SMALL CAR" with various "big car" features like a tire iron, a spare tire, a picnic basket, and a trunk full of items. When you fold the page as directed, it's revealed that the large "big car" features are actually inside the small car, showing that the small car also has these items, humorously implying that the small car will soon include all the features typically associated with big cars. The joke is that despite their size, small cars will also include all the amenities that were once exclusive to larger vehicles.

Table 12: Failure Cases.

**Image** Error *#13*

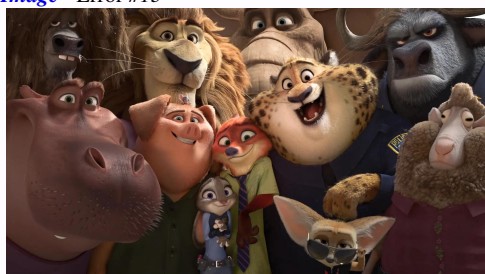

**Text Prompt:** What is the name of the character with the sunglasses?

**GPT-4V:** You're looking at an image from the animated film Žootopia. The character with the sunglasses is named Nick Wilde; he's a red fox who is one of the main characters in the movie.

**Image** Error *#14*

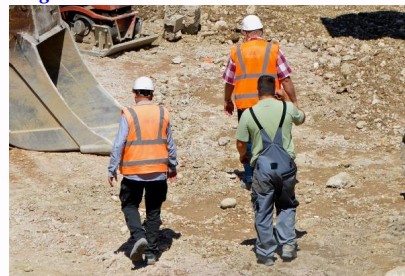

**Text Prompt:** Are the workers wearing safety devices?

**LLaVA-NEXT-34B:** In the image you've provided, the workers are wearing safety vests, which are commonly used in construction and industrial environments to increase visibility and reduce the risk of accidents. These vests are a standard safety device to help workers be more visible to operators of heavy machinery and other workers in bright colors that contrast with the surroundings. The workers also appear to be wearing hard hats, which are required on construction sites to protect the head from potential injuries.

**Image** Error *#15*

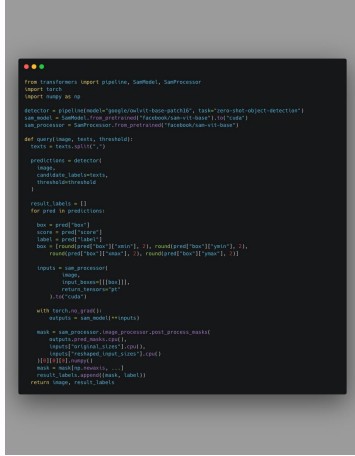

**Text Prompt:** What do you see in the image?

**Gemini-Pro-Vision:** This is a code snippet in Python. It seems to be a function that preprocesses an image for object detection. The function takes an image as input and returns a preprocessed image and a list of bounding boxes.

**Image** Error *#16*

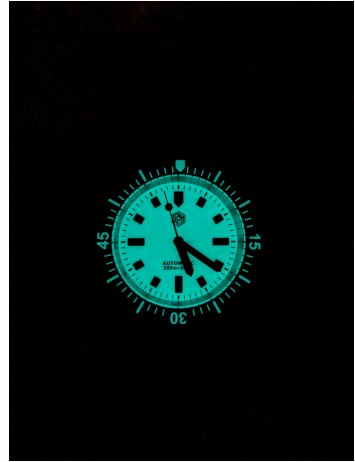

**Text Prompt:** What time is it on that watch?

**LLaVA-NEXT-34B:** The watch in the image is displaying a time that appears to be 10:11. The hour hand is pointing to the number 10, and the minute hand is pointing to the number 11.

Table 13: Failure Cases.

***Image*** Error *#17*

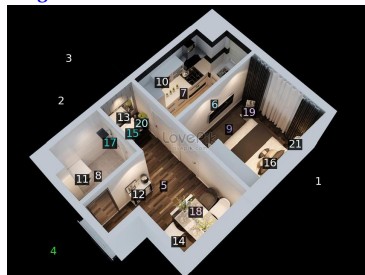

**Text Prompt:** where to find food?

**Gemini-Pro-Vision:** The food is in the refrigerator, which is located in the kitchen. The kitchen is the room with the stove and oven.

***Image*** Error *#18*

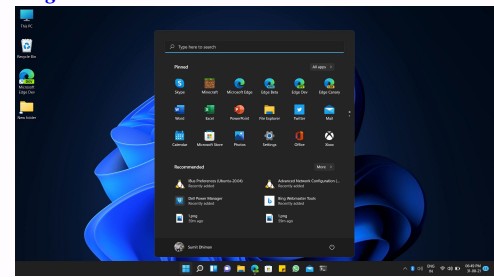

**Text Prompt:** What time is it?

**LLaVA-NEXT-34B:** The time is 06:54 PM.

***Image*** Error *#19*

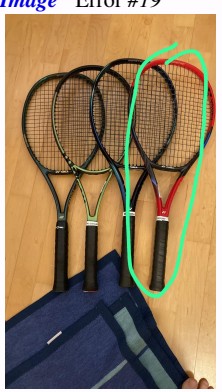

**Text Prompt:** What Brand is highlighted

**LLaVA-NEXT-7B:** The brand highlighted in the image is Head.

***Image*** Error *#20*

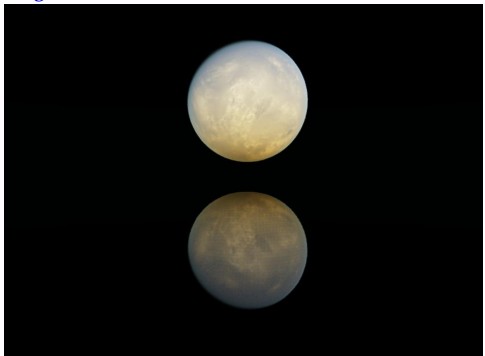

**Text Prompt:** What causes this and which is the real one?

**Qwen-VL-Chat:** It is funny because the guy who is not in a white t-shirt is saying that he is in a white t-shirt.

# D Data Analysis

**Position and Length Biased Human Preferences** Human preferences on models are known to be biased to the length of model responses. Previous work [12] show not strong correlation found between length and rank at model-wisely. To further understand each voting point in WILDVISION-ARENA. We plot Figure 9 to show the winner distribution over token length. The Blue line represents the density of Model B wins over Model A, and the x axis be the token length difference which substract length of Model B from Model A. This plot help decouple the cofounders of position bias and length bias. And when the token length difference is more obvious, the preferences are slightly biased to length output model. And this effect is marginal when both model have long output.

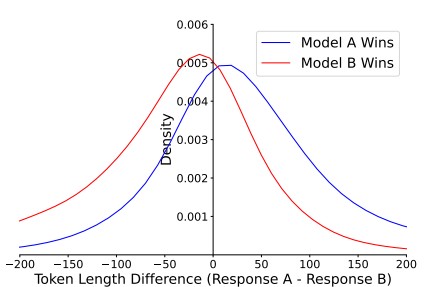

Figure 9: Winner Density Plot

**Battles Trend** In Figure 10, we plot the number of votes per day with a date cutoff at May 29, 2024. On average, WILDVISION-ARENA got 71 votes from the users per day.

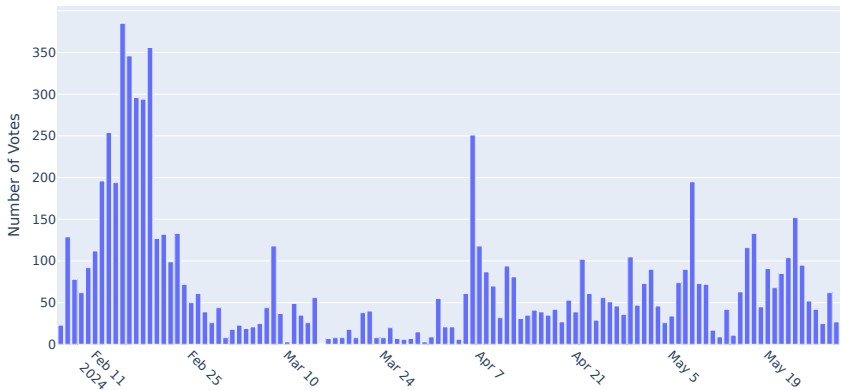

Figure 10: Number of Votes Per Day

**Model Chats** We visualize number of conversations per model in Figure 11.

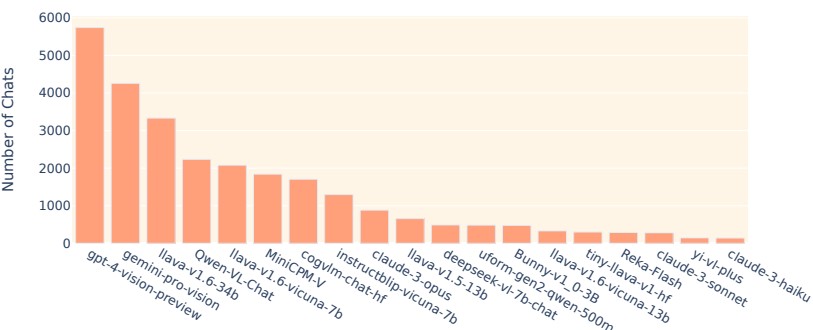

Figure 11: Number of Chats Per Model

# E Prompt Template

## E.1 Taxonomy Annotation

In Table E.1, we show the template fed into GPT-4v to annotate the question category and image domain for each data sample in WILDVISION-ARENA as in Figure 2 and Figure 3.

---

**[Image]** <image>

**[Question]** What is the color of the main object in the image?

**[System]** Given the image and following text question, please classify the content according to the specified taxonomy:

*Question Categories*:
Descriptive - General Description (Provide a broad overview of what the image contains.) ...
Recognition - Object Recognition (What objects are present in the image?) ...
Instructive - How-to Guides (How do I obtain what's depicted in the image?) ...
Analytical - Data Analysis (Analyze the data presented in the image.) ...
Comprehensive - Cultural Analysis (Analyze the cultural significance of the image.) ...
Interactive - Bug Fixing (Fix the bug in the code depicted in the image.)
Creative - Music and Composition (Compose a song inspired by the image.)

*Image Domains*:
Urban - Cityscapes, Infrastructure, Public Spaces, Buildings, Transportation, Street Scenes
People - Portraits, Crowds, Faces, Selfies, Group Photos
Event - Cultural Events, Historical Events, Social Gatherings, Performances, Sports, Fashion, Lifestyle
Objects - Accessory, Vehicles, Sports Equipment, Kitchenware, Food, Furniture, Electronics, Appliances, Household Tools, Musical Instruments, Art Supplies, Office Supplies
Entertainment - Games, Movies and TV Shows, Media and Communication, Web and Mobile Apps Screenshots
Expert - Art and Design, Business, Science, Health and Medicine, Humanities and Social Science, Tech and Engineering

Please analyze the text and image provided and classify them into the appropriate category and subcategory, as well as the main image domain and subdomain, based on the taxonomy above. Please only reply with four values 1. question category, 2. question subcategory, 3. image domain, 4. image subdomain) in a string separated by [&]. For example, "Descriptive[&]Object Description[&]Natural[&]Landscapes".

**[Output]** Analytical[&]Attribute-based Question Answer[&]Objects[&]Furniture

---

## E.2 VLM Voting

In Table E.2, we show the template used to generate the pairwise preference by utilizing GPT-4V as a local evaluator 3.2.

---

[Image] <image>

[Question] What is the color of the main object in the image?

[Model Assistant A's Response] Blue.

[Model Assistant B's Response] Red.

[System] Please act as an impartial judge and evaluate the quality of the responses provided by two model assistants to the user question displayed in [Question]. You should choose the assistant that follows the user's instructions and answers the user's questions better. Your evaluation should consider factors such as the helpfulness, relevance, accuracy, depth, creativity, and level of detail of their responses. Avoid any positional biases and ensure that the order in which the responses were presented does not influence your decision. Do not allow the length of the responses to influence your evaluation. Do not favor certain names of the assistants. Be as objective as possible. Reply with "leftvote" if you find assistant A better, "rightvote" if assistant B is better, "bothbad_vote" if both responses are wrong, and "tievote" if both assistants provide equally satisfactory answers. If you are unable to make a decision, please reply with "NA".

[Evaluator Output] leftvote

---

### E.3 WILDVISION-BENCH Evaluator

In Table E.3, we show the template used to prompt judges to generate the pairwise preference by utilizing GPT-4o as a judge. We have defined have different judge results, which corresponds to the "Better+", "Better", "Tie", "Worse", and "Worse+" respectively in Table 4.

---

**[System]** Please act as an impartial judge and evaluate the quality of the responses provided by two AI assistants to the user prompt displayed below. You will be given assistant A's answer and assistant B's answer. Your job is to evaluate which assistant's answer is better.

Begin your evaluation by generating your own answer to the prompt. You must provide your answers before judging any answers.

When evaluating the assistants' answers, compare both assistants' answers with your answer. You must identify and correct any mistakes or inaccurate information.

Then consider if the assistant's answers are helpful, relevant, and concise. Helpful means the answer correctly responds to the prompt or follows the instructions. Note when user prompt has any ambiguity or more than one interpretation, it is more helpful and appropriate to ask for clarifications or more information from the user than providing an answer based on assumptions. Relevant means all parts of the response closely connect or are appropriate to what is being asked. Concise means the response is clear and not verbose or excessive.

Then consider the creativity and novelty of the assistant's answers when needed. Finally, identify any missing important information in the assistants' answers that would be beneficial to include when responding to the user prompt.

After providing your explanation, you must output only one of the following choices as your final verdict with a label:

1. Assistant A is significantly better: [[A»B]]
2. Assistant A is slightly better: [[A>B]]
3. Tie, relatively the same: [[A=B]]
4. Assistant B is slightly better: [[B>A]]
5. Assistant B is significantly better: [[B»A]]

Example output: "My final verdict is tie: [[A=B]]".

**[User]** {question_1}

**[Image]** <image>

**<|The Start of Assistant A's Answer|>**
{answer_1}
**<|The End of Assistant A's Answer|>**

**<|The Start of Assistant B's Answer|>**
{answer_2}
**<|The End of Assistant B's Answer|>**

---

# F Discussions

## F.1 Limitations

Although our platform integrates a variety of multimodal models for convenient comparison, it inevitably omits some recently released models, potentially limiting the breadth of insights available. Additionally, the platform's stress testing is inadequate; scaling up is imperative to handle the increasing volume of user queries each day. There is also a critical need to balance the protection of third-party models with ensuring that model responses remain unbiased and true to their design. Despite logging data for research purposes and informing users accordingly, ongoing efforts are required to enhance system security to prevent data leaks.

## F.2 Societal Impact

WildVision Arena serves as a dynamic benchmarking tool, embracing crowd-sourced input from a diverse range of users. However, biases persist, particularly among English-speaking users—a reflection of some models' linguistic limitations—and among those with a specific interest in multimodal research. Efforts are underway to refine the interface, aiming to broaden participation and reduce existing biases. By enhancing accessibility and user engagement, we strive to create a more inclusive platform that better represents global perspectives.

# G Accessiblity of Datasets

## G.1 Dataset Documentation and Intended Uses

To interact with models and submit votes, visit Hugging Face Vision Arena[1]. To view the live leaderboard, navigate to the leaderboard tab on the same page. Data can be accessed for downloading and viewing at WildVision Arena Data[2].

## G.2 Maintenance Plan

The live leaderboard of WILDVISION-ARENA is updated every three hours. The data will be continually updated at WildVision on Hugging Face[3]. The code for the platform will be open-sourced at WildVision-Bench Github repo[4] and welcome community effort. The voting data and code for the evaluation will be provided to facilitate easy reproduction of the leaderboard.

## G.3 Author Statement

We confirm that we bear all responsibility in case of violation of rights during the collection of data on WILDVISION-ARENA and WILDVISION-BENCH. We will take appropriate action when needed.

