**Part I**

# Appendix

## Table of Contents

# A  User Interface

In Figure 8, we show a screenshot of the user interface of our WILDVISION-ARENA, which presents an interactive environment for evaluating multimodal large language models. This environment allows users to input questions and compare responses from multiple models simultaneously. Each model's answer is displayed side-by-side, enabling a straightforward comparison of their performance and capabilities based on user queries related to specific images or tasks. The interface also facilitates easy selection and voting to decide which model's response fits the user's criteria best, enhancing the user's ability to judge and refine the models' outputs effectively.

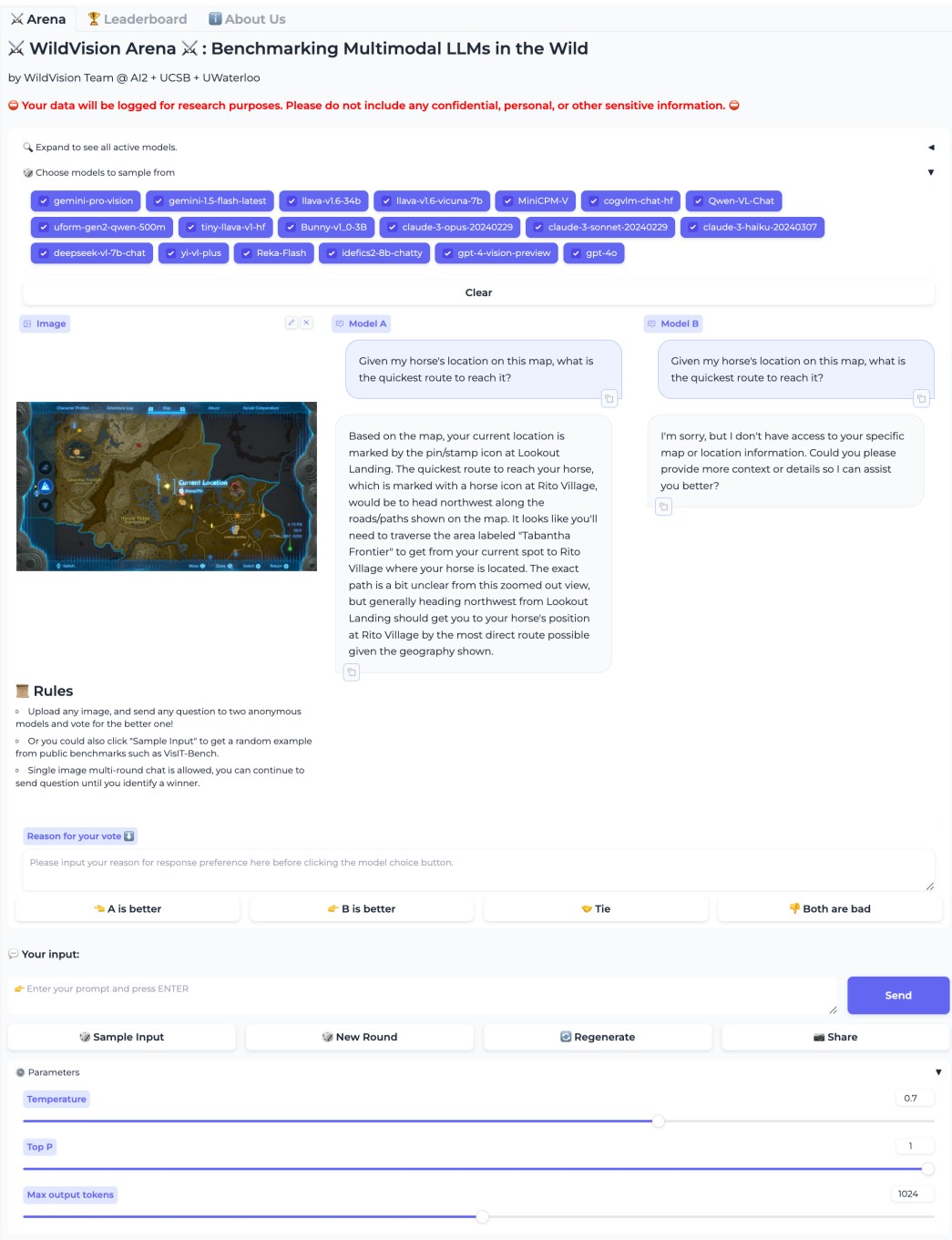

Figure 8: User Interface of WILDVISION-ARENA.

# B  Question Category and Image Domain

In Table 6- 8, we showcase example data under each of the image domain and question category from WILDVISION-ARENA's users.

Table 6: Example input data in WILDVISION-ARENA tagged with [Image Domain-Subdomain] and [ Question Category-Subcategory].

*Image* [Entertainment-Movies/TV Shows]

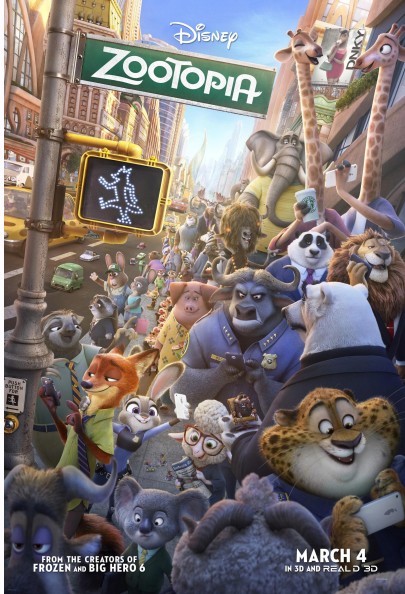

[Descriptive-Movies/TV Shows] **Text Prompt:** What are the two giraffe characters on this movie poster doing?

*Image* [Natural-Plants]

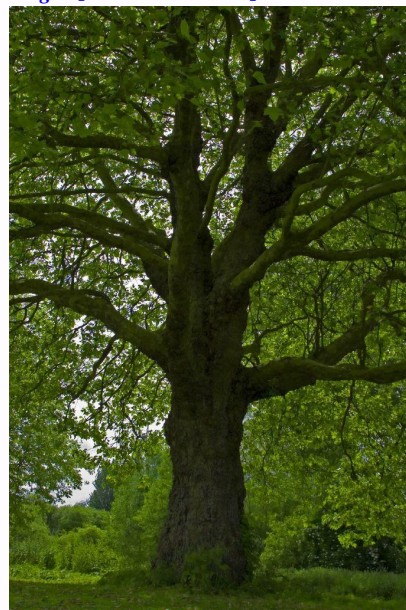

[Analytical-Problem Solving] **Text Prompt:** How likely is it to snow after this picture was taken? What would change with this type of tree before it's likely to snow?

*Image* [Expert-Business]

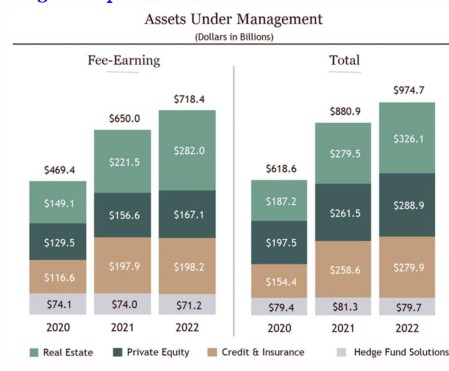

[Analytical-Data Analysis] **Text Prompt:** Which of the companies featured in the dashboard are headquartered outside the US?

*Image* [Urban-Infrastructure]

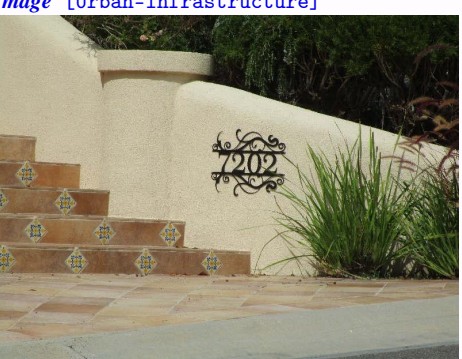

[Recognition-Text] **Text Prompt:** Can you tell me the potential risks and the unreasonale parts in the image?

Table 7: Example input data in WILDVISION-ARENA tagged with [Image Domain-Subdomain] and [ Question Category-Subcategory].

*Image* [Entertainment-Comics]

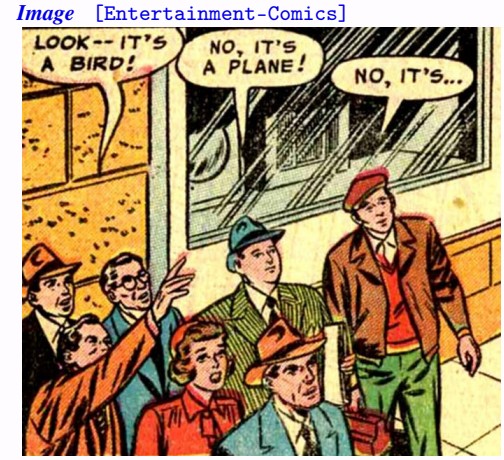

[Descriptive-Scene Description] **Text Prompt:** Whos's in the sky?

*Image* [People-Portraits]

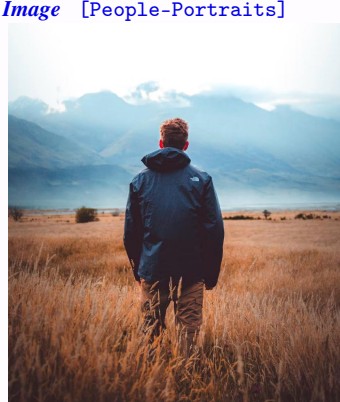

[Creative-Media Post] **Text Prompt:** write a social media post with the provided image, saying that I am ready for the new challange.

*Image* [Urban-Buildings]

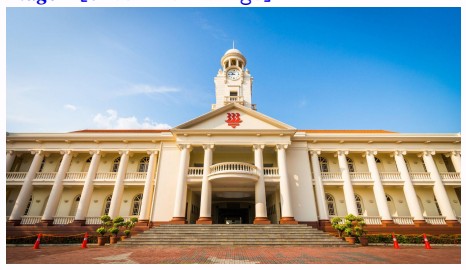

[Recognition-Location] **Text Prompt:** where is this?

*Image* [Expert-Science]

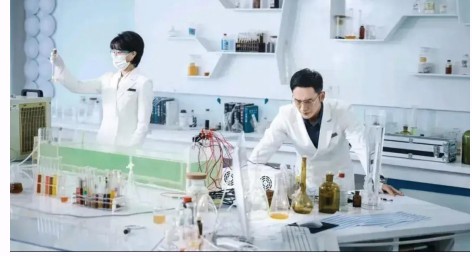

[Analytical-Safety Procedures] **Text Prompt:** Can you tell me the potential risks and the unreasonale parts in the image?

*Image* [Natural-Landscapes]

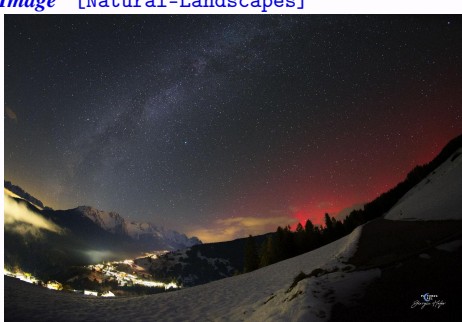

[Recognition-Location] **Text Prompt:** where was this photo taken?

*Image* [Objects-Household Tools]

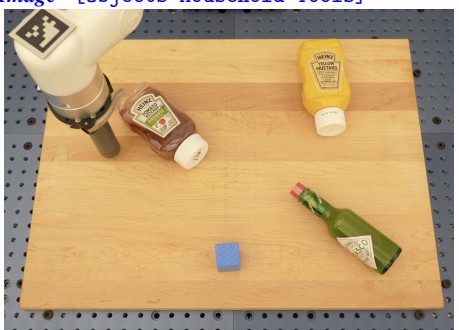

[Descriptive-Object Description] **Text Prompt:** describe the scene and objects

Table 8: Example input data in WILDVISION-ARENA tagged with [Image Domain-Subdomain] and [ Question Category-Subcategory].

*Image* [Entertainment-Web and Mobile Apps Screenshots]

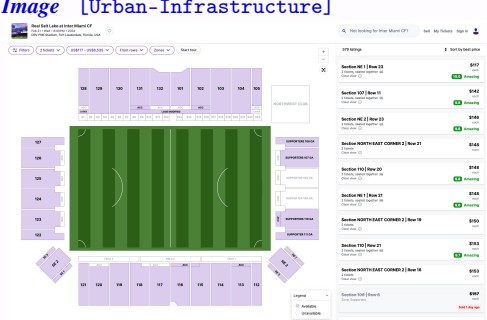

[Interactive-Web Navigation] **Text Prompt:** I need to download flyer, you will be given screenshot from browser with elements marked with number. give next action to take on web page to download the flyersngive me response in below format example 1 action:[click,scroll,wait], box:1 format action:, box:

*Image* [Event-Sports]

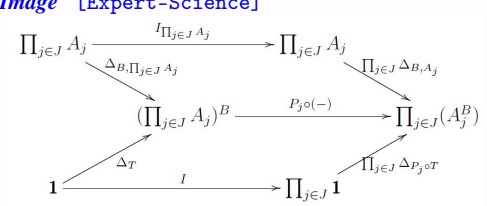

[Descriptive-Scene Description] **Text Prompt:** this is a football match , every player has an identifier , describe every player action (example : player #501 is running)

*Image* [Urban-Infrastructure]

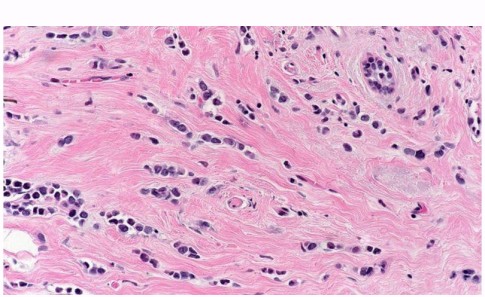

[Interactive-Recommendations] **Text Prompt:** Which section's ticket would you recommend I purchase?

*Image* [Expert-Science]

$$\begin{array}{ccc} \prod_{j \in J} A_j & \xrightarrow{I_{\prod_{j \in J} A_j}} & \prod_{j \in J} A_j \\ {\scriptstyle \Delta_{B, \prod_{j \in J} A_j}} \searrow & & \downarrow {\scriptstyle \prod_{j \in J} \Delta_{B, A_j}} \\ & (\prod_{j \in J} A_j)^B \xrightarrow{P_j \circ (-)} \prod_{j \in J} (A_j^B) \\ {\scriptstyle \Delta_T} \nearrow & & \nearrow {\scriptstyle \prod_{j \in J} \Delta_{P_j \circ T}} \\ 1 & \xrightarrow{I} & \prod_{j \in J} 1 \end{array}$$

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

---

[1]https://huggingface.co/spaces/WildVision/vision-arena

[2]https://huggingface.co/datasets/WildVision/wildvision-arena-data

[3]https://huggingface.co/WildVision

[4]https://github.com/WildVision-AI