# OpenReview forum: "WildVision: Evaluating Vision-Language Models in the Wild with Human Preferences"
_NeurIPS.cc/2024/Datasets_and_Benchmarks_Track — NeurIPS 2024 Track Datasets and Benchmarks Poster_

### Official Review · Reviewer_WVHU · 2024-07-18

**Rating:** 6
**Confidence:** 4
**Correctness:** Yes.
**Clarity:** Yes.

**Review:**

The paper studies human preference within interactions with VLMs. This is an interesting direction. Through the developed online platform WV-Arena, it uses GPT-4 as the judge to compare each VLM with Claude-3-Sonnet, showing a higher Spearman correlation with the WV-Arena Elo rating system than other benchmarks such as MMVet, MMMU, and MMStar. It delivers comprehensive analysis showing multiple failure cases of top-performing VLMs, e.g., they face challenges with subtle contextual cues, spatial reasoning, visual imagination, and expert domain knowledge. VLMs also exhibit issues with hallucinations and safety when intentionally provoked. Below are some questions explaining how this paper can be further improved or revised.

- VLMs are indeed powerful foundation models that can yield impressive results, yet the studied VLMs like GPT-4o and GPT-4V wrap VLMs using various engineering components which are even unknown to common users. That said, the output by these models are post-processed towards more user-friendly experience. Their better performance than others is not simply owing to proprietary data (L126). Therefore, it is not clear how these post-processing engineering add-ons affect the conclusions. Authors are suggested to discuss this point.

- The paper reports a high Spearman correlation of VLMs' output w.r.t real human preference. While this is a promising thing, it is interesting to see in what cases VLMs' output do not align with human preference. For example, the paper proposes to use GPT-4 as the judge to compare each VLM with Claude-3-Sonnet, achieving high Spearman correlation 0.94. But what cases make the correlation not perfect? Authors are suggested to have a discussion so that the reader can understand the limitations of the developed method.

- The WV-Bench benchmark exploits GPT-4V which is a commercial product. As far as I know, it requires an account and limits eligibility of global users. Therefore, it is not clear whether this benchmark is accessible to global practitioners and researchers. It is also not clear how the evaluation speed is, whether there is quota in the evaluation (because of using commercial software). Authors should discuss.

- To my understanding, users upload images for which the authors had no control. If so, it is not clear whether these images have copyrights and whether using them in the benchmark has copyright issues. Authors should discuss this.

**Strengths:**

- It is interesting to study how VLMs' performance aligns with human preference.
- The paper makes a good effort to set up the study by developing an online platform and curating a benchmark dataset.
- The writing and organization of the paper are good, containing sufficient details and benchmarking enough representative and state-of-the-art VLMs.

**Additional Feedback:**

Authors are encouraged to refer to previous boxes to submit rebuttal. Rating will be updated according to the rebuttal.

**Documentation:**

Yes.

**Ethics:**

To my understanding, when authors collect data through the developed platform, they let users upload images for which the authors had no control. Then, the curated benchmark WV-Bench selects images from user-uploaded images (along with VLMs' results and human preference recordings). If so, it is not clear whether these images have copyrights and whether using them in the benchmark has copyright issues. There might be copyright issues and Ethics review might be needed.

**Limitations:**

Authors discuss limitations in the supplementary material but not the main paper. Authors are suggested to move this discussion to the main text. Moreover, authors can refer to the Review box to include more discussions on limitations, e.g., accessibility of the proposed benchmark, discounting engineering add-ons in commercial platforms that wrap VLMs, etc.

**Opportunities For Improvement:**

Authors are encouraged to refer to Review box for the questions. Authors can revise and submit rebuttal. The rating will be updated according to the rebuttal and revision.

**Relation To Prior Work:**

Yes.

**Summary And Contributions:**

The paper aims to benchmark vision-language models (VLMs) in terms how their performance aligns with human preferences. Towards this goal, it launched WildVision-Arena (WV-Arena), an online platform that collected human preferences to evaluate VLMs. Comprehensive anlaysis of the data leads to rich observations and conclusions. The paper summarizes three contributions:
- developing the platform WildVision-Arena to collect data to study human preferences in interaction with different VLMs,
- curating WildVision-Bench from WV-Arean, a benchmark to report how VLMs align with human preference,
- releasing 20,000+ multimodal conversations and 8,000+ votes to advance VLM research.

---

> ### Author Response · Authors · 2024-08-18
> **Response to Reviewer WVHU**
>
> Thank you for your insightful feedback. We greatly appreciate your recognition of the novelty of our study and the value of our platform, as well as the dataset. We would like to address your concerns as follows:
>
>
>
> ### Q1: Discussion of adds-on effect of Close-source VLMs
>
>
> Thank you for highlighting this important issue. We agree that the implementation details of closed-source models like GPT-4 and GPT-4V have not been disclosed to the public.
>
> Given that the specifics of these models are not available, we are unable to perform ablation studies to analyze the impact of "post-processing engineering add-ons" because we don't have access to their original, unprocessed outputs.
>
> In our online leaderboard, we have added a special marker to differentiate between open and closed models. The significant gap between them could be attributed to factors such as: (i) proprietary data, (ii) larger model size, (iii) engineering components including post-processing, and more. Our WV-Arena platform allows for pairwise comparisons, making it possible to identify these contributing factors and ultimately help in building a more advanced VLM model or system. We will include this detailed discussion at the beginning of Section 3.
>
>
> ---
>
> ### Q2: What cases make the correlation not perfect
>
>
> We carefully reviewed the cases where the judgments did not align with expert opinions. To summarize, there are three main scenarios:
>
> 1. The tasks are challenging even for the judge model.
> 2. Both model responses address the answer to some extent, making the evaluation somewhat subjective.
> 3. GPT-4, as a judge, tends to favor GPT-aligned models more than human raters, such as favoring the LLaVA model more than human evaluators.
>
> Despite these limitations of GPT-4, it remains one of the most reliable automatic judge models currently available to us.
>
>
> ---
>
> ### Q3: Using GPT-4V models as judges
>
> We understand the concern that GPT-4V and GPT-4O are not freely available for evaluation, and we have addressed this in the following ways:
>
> - **Free Credits:** We offer free credits to model authors who wish to submit their models to our leaderboard. This means they won't need to pay anything to OpenAI for evaluations.
>
> - **Using Open VLMs as Judges:** We will add a new column to display results using the best open-source VLMs as judges, specifically those with the highest correlation to human preferences. This allows users who do not have access to GPT models to evaluate their models using open-weight VLMs, such as the largest versions of Qwen-VL and Llava-Next.
>
> We also want to emphasize that using the GPT-4 series as a judge is widely recognized in both LLM evaluation ([1][2]) and VLM evaluation ([3]) for open-ended tasks. We will incorporate this discussion in Section 3.2, lines 147-155.
>
>
> ---
>
>
> ### Q4: Copyrights of the data in Benchmarks
>
> To clarify, in our publicly released dataset, WV-Bench, all 500 examples have undergone multiple rounds of human verification, and we are confident that they do not contain any copyright or safety issues. For future releases of our data and updates to the benchmark, we will continue to use both automatic annotation and human annotators to identify and remove any data that may have copyright concerns.
>
> > I played the huggingface interface and I find that there is not guarantee to avoid copyright issue
>
> Currently, our moderation is conducted in a post-processing manner. We do not restrict authors from uploading or querying images that may have copyright concerns at the initial stage. However, before releasing any data to the public, we carry out a thorough filtering and annotation process. This process involves human annotators, making it difficult to implement directly within the user interface itself.
>
>
> Here is the link to our public data: https://huggingface.co/datasets/WildVision/wildvision-bench
>
>
>
> ---
>
> [1] Zheng, L., et al. (2023). Judging LLM-as-a-Judge with MT-Bench and Chatbot Arena. Proceedings of the Neural Information Processing Systems Conference (NeurIPS 2023).
>
> [2] Yann, D., et al. (2024) Length-Controlled AlpacaEval: A Simple Way to Debias Automatic Evaluators.
>
> [3] Yu, J., et al. (2024). MM-Vet: Evaluating Large Multimodal Models for Integrated Capabilities. Proceedings of the International Conference on Machine Learning (ICML 2024).
>
>
> ** Discussion of Limitations in the Main Paper**
> Thank you for your suggestions. We will revise our paper accordingly to discuss these limitations in more detail in the main text, taking advantage of the additional space available in the camera-ready version.
>
> We sincerely hope that these clarifications address your concerns. If so, we kindly request that you consider adjusting your score accordingly. Thank you once again for your valuable feedback.

---

> > ### Comment · Reviewer_WVHU · 2024-08-19
> >
> > Thanks for the rebuttal. I read all the reviews and the rebuttal. The rebuttal does not effectively address my concerns. I have several points of concerns but the authors merge my questions and provide general response. This makes it hard for me to find the answers by myself. For example, authors state that "Regarding the accessibility of the benchmark, ... Details on this can be found in Supplemental Materials G". However, I do not find relevant information about how to ensure general users have accessibility to GPT-4V, which is a cornerstone in the proposed benchmark.
> >
> > For the copyright issue, frankly, I am not convinced the proposed solutions can address this issue; I played the huggingface interface and I find that there is not guarantee to avoid copyright issues. Anyways, I'll defer to Ethics Reviewers decide how to move on.

---

> > > ### Author Rebuttal · Authors · 2024-08-21
> > >
> > > Thank you again for your effort in bringing up these questions and suggestions. We greatly appreciate your valuable feedback and the time you put into the review process.
> > >
> > > We apologize for the mistake in our previous message, where our responses to the four questions were accidentally left out during the copy-paste process. We have now attached the responses here and have also updated the previous message to include them.
> > >
> > > ---
> > >
> > >
> > > ### Q1: Discussion of adds-on effect of Close-source VLMs
> > >
> > >
> > > Thank you for highlighting this important issue. We agree that the implementation details of closed-source models like GPT-4 and GPT-4V have not been disclosed to the public.
> > >
> > > Given that the specifics of these models are not available, we are unable to perform ablation studies to analyze the impact of "post-processing engineering add-ons" because we don't have access to their original, unprocessed outputs.
> > >
> > > In our online leaderboard, we have added a special marker to differentiate between open and closed models. The significant gap between them could be attributed to factors such as: (i) proprietary data, (ii) larger model size, (iii) engineering components including post-processing, and more. Our WV-Arena platform allows for pairwise comparisons, making it possible to identify these contributing factors and ultimately help in building a more advanced VLM model or system. We will include this detailed discussion at the beginning of Section 3.
> > >
> > >
> > > ---
> > >
> > > ### Q2: What cases make the correlation not perfect
> > >
> > >
> > > We carefully reviewed the cases where the judgments did not align with expert opinions. To summarize, there are three main scenarios:
> > >
> > > 1. The tasks are challenging even for the judge model.
> > > 2. Both model responses address the answer to some extent, making the evaluation somewhat subjective.
> > > 3. GPT-4, as a judge, tends to favor GPT-aligned models more than human raters, such as favoring the LLaVA model more than human evaluators.
> > >
> > > Despite these limitations of GPT-4, it remains one of the most reliable automatic judge models currently available to us.
> > >
> > >
> > > ---
> > >
> > > ### Q3: Using GPT-4V models as judges
> > >
> > > We understand the concern that GPT-4V and GPT-4O are not freely available for evaluation, and we have addressed this in the following ways:
> > >
> > > - **Free Credits:** We offer free credits to model authors who wish to submit their models to our leaderboard. This means they won't need to pay anything to OpenAI for evaluations.
> > >
> > > - **Using Open VLMs as Judges:** We will add a new column to display results using the best open-source VLMs as judges, specifically those with the highest correlation to human preferences. This allows users who do not have access to GPT models to evaluate their models using open-weight VLMs, such as the largest versions of Qwen-VL and Llava-Next.
> > >
> > > We also want to emphasize that using the GPT-4 series as a judge is widely recognized in both LLM evaluation ([1][2]) and VLM evaluation ([3]) for open-ended tasks. We will incorporate this discussion in Section 3.2, lines 147-155.
> > >
> > >
> > > ---
> > >
> > >
> > > ### Q4: Copyrights of the data in Benchmarks
> > >
> > > To clarify, in our publicly released dataset, WV-Bench, all 500 examples have undergone multiple rounds of human verification, and we are confident that they do not contain any copyright or safety issues. For future releases of our data and updates to the benchmark, we will continue to use both automatic annotation and human annotators to identify and remove any data that may have copyright concerns.
> > >
> > > > I played the huggingface interface and I find that there is not guarantee to avoid copyright issue
> > >
> > > Currently, our moderation is conducted in a post-processing manner. We do not restrict authors from uploading or querying images that may have copyright concerns at the initial stage. However, before releasing any data to the public, we carry out a thorough filtering and annotation process. This process involves human annotators, making it difficult to implement directly within the user interface itself.
> > >
> > >
> > > Here is the link to our public data: https://huggingface.co/datasets/WildVision/wildvision-bench
> > >
> > >
> > >
> > > ---
> > >
> > > [1] Zheng, L., et al. (2023). Judging LLM-as-a-Judge with MT-Bench and Chatbot Arena. Proceedings of the Neural Information Processing Systems Conference (NeurIPS 2023).
> > >
> > > [2] Yann, D., et al. (2024) Length-Controlled AlpacaEval: A Simple Way to Debias Automatic Evaluators.
> > >
> > > [3] Yu, J., et al. (2024). MM-Vet: Evaluating Large Multimodal Models for Integrated Capabilities. Proceedings of the International Conference on Machine Learning (ICML 2024).
> > >
> > > ---
> > >
> > > ### Thank you!
> > >
> > > We apologize once again for the oversight in our previous message, where we mistakenly omitted our responses to the four questions. We appreciate your understanding and have now ensured that the responses are included. If you have any further concerns or follow-up questions, please don’t hesitate to reach out. Thank you for your continued patience and support.

---

> > > > ### Comment · Reviewer_WVHU · 2024-08-30
> > > >
> > > > Thanks for the follow-up responses, which address my concerns to some extent. I defer the evaluation of ethics and copyright to Ethics Reviewers. I'll maintain my rating and strongly suggest that authors include the discussions (presented in the rebuttal) in the paper to improve its quality.

---

> > > > > ### Author Response · Authors · 2024-08-31
> > > > >
> > > > > Thank you for your feedback. We will definitely include the discussions from our rebuttal in the main paper to improve its quality. Your suggestions are much appreciated.

---

> ### Author Response · Authors · 2024-08-28
>
> Dear Reviewer,
>
> Thank you for your valuable feedback, which has significantly improved our paper. We have addressed your concerns in our rebuttal and hope our latest responses meet your expectations. As the discussions window is ending, please let us know if you have further questions. We appreciate your consideration and kindly request you to adjust your scores accordingly if satisfied with our clarifications.

---

### Official Review · Reviewer_dzdh · 2024-07-21

**Rating:** 4
**Confidence:** 5
**Correctness:** Yes
**Clarity:** No

**Review:**

This paper focuses on how to benchmark human preferences in real-world multimodal interactions for vision-language models. Specifically, the authors developed an online platform WV-ARENA that collects human preferences. Based on the platform, the authors constructed a benchmark dataset WV-Bench that contains 500 samples. Some research findings about different VLMs on the benchmark dataset are provided.

Pros:
1) This paper focuses on how to benchmark human preferences for vision-language models. This is an important yet understudied problem.
2) The evaluation results using the proposed benchmark achieve a good Spearman correlation, show the benchmark data aligns with human preferences.
3) Some interesting fail cases of existing models are provided and discussed.

Cons:
1) The datasets only contain 500 samples, which is quite small scale.
2) The construction of the dataset is not clear and should be described in more detail. For example, how to guarantee the safety and diversity of the dataset, more statistical results and illustration examples should be provided.
3) The authors adopt the GPT-4O as a judge model. It would be better to do a human judge.
4) The link to the platform WV-ARENA is not given in the paper. Is it not available?

**Strengths:**

Pros:
1) This paper focuses on how to benchmark human preferences for vision-language models. This is an important yet understudied problem.
2) The evaluation results using the proposed benchmark achieve a good Spearman correlation, show the benchmark data aligns with human preferences.
3) Some interesting fail cases of existing models are provided and discussed.

**Additional Feedback:**

No

**Documentation:**

No

**Limitations:**

Yes

**Opportunities For Improvement:**

Cons:
1) The datasets only contain 500 samples, which is quite small scale.
2) The construction of the dataset is not clear and should be described in more detail. For example, how to guarantee the safety and diversity of the dataset, more statistical results and illustration examples should be provided.
3) The authors adopt the GPT-4O as a judge model. It would be better to do a human judge.
4) The link to the platform WV-ARENA is not given in the paper. Is it not available?

**Relation To Prior Work:**

Yes

**Summary And Contributions:**

This paper focuses on how to benchmark human preferences in real-world multimodal interactions for vision-language models. Specifically, the authors developed an online platform WV-ARENA that collects human preferences. Based on the platform, the authors constructed a benchmark dataset WV-Bench that contains 500 samples. Some research findings about different VLMs on the benchmark dataset are provided.

---

> ### Author Response · Authors · 2024-08-18
> **Response to Reviewer dzdh**
>
> Thank you for your thoughtful and constructive feedback on our paper. We appreciate your recognition of the importance of our work in addressing this understudied problem and the value of our benchmark and analysis. We believe that the points you raised provide valuable opportunities to further improve our work, and we would like to address your concerns in detail.
>
> **1. Scale of Datasets**
>
> We understand your concern about the dataset size. However, we respectfully argue that the scale of our WV-Bench is both reasonable and appropriate within the current research landscape. For instance, the MT-Bench [1] contains 80 questions, and the Arena-Hard benchmark [2], which evaluates large language models (LLMs) in real-world scenarios, also consists of 500 samples. RealworldQA [3], a benchmark dataset released by xAI for evaluating vision-language models (VLMs) in real-world settings, comprises 765 test samples. Additionally, a widely-used VLM benchmark MM-Vet also only has 218 samples.
>
> Our dataset, while consisting of 500 samples, is derived from a larger pool of over 20,000 chat instances and 8,000+ battle data points (as of the submission date). The selected 500 samples are highly diversified and exhibit a strong correlation with the broader dataset, confirming their representativeness. Moreover, our benchmark will be regularly updated as the volume of collected data continues to grow, ensuring its continued relevance and robustness.
>
> **2. Details of Ensuring Safety and Diversity**
>
> We take the safety and diversity of our dataset very seriously. To ensure the safety of the content, we employ an NSFW detector to filter out any unsafe images. Additionally, our team of experts manually reviews the examples to further eliminate any unsafe image or text samples. Regarding diversity, we sequentially add samples to the pool and filter out those with duplicated images or questions, as well as those with a CLIP similarity score of over 0.7 in image features compared to any sample already in the pool. These details will be explicitly included in Section 3.1 of our paper, titled "Data Curation Pipeline," to provide clarity on our processes.
>
> **3. Human Judge**
>
> We would like to clarify that human judgments are indeed a central component of our evaluation process. The human judgments you mentioned are crowdsourced through our WV-Arena platform, which is a core part of our contribution. As presented in Table 2, these judgments are provided by real-world users across a wide range of tasks, and the votes collected are used to estimate Elo ratings using the Bradley-Terry model, as described in Section 2.4.
>
> In addition to these human judgments, we also use GPT-4O as a judge model for our test split. While this serves as a fast evaluation method, it also demonstrates a high correlation with human judgments, as depicted in Figure 7. This dual approach of human and model-based evaluations ensures that our benchmark is both robust and efficient.
>
> **4. Link to the Platform WV-ARENA**
>
> The link to the WV-Arena platform is indeed provided on the first page of the main paper, directly below the list of authors. Additionally, the link is included in Section G.1 of the Supplemental Materials. We invite you to explore our platform, and we welcome any further suggestions you may have for its improvement.
>
> **Request for Score Reconsideration**
>
> Given the clarifications and additional details provided above, we hope that our responses address your concerns effectively. We believe that our work makes a significant contribution to the field and that the revisions we plan to implement will further strengthen the paper. We kindly request that you consider raising your score if you find our explanations satisfactory.
>
> Thank you once again for your detailed review and for your contributions to improving our work. We sincerely appreciate your time and effort.
>
> [1] Zheng, L., et al. (2023). Judging LLM-as-a-Judge with MT-Bench and Chatbot Arena. Proceedings of the Neural Information Processing Systems Conference (NeurIPS 2023).
>
> [2] Li, Z., et al. (2024). From Crowdsourced Data to High-Quality Benchmarks: Arena-Hard and BenchBuilder Pipeline. Under Review.
>
> [3] xAI. (2024). RealworldQA. Dataset released by xAI.
>
> [4] Yu, J., et al. (2024). MM-Vet: Evaluating Large Multimodal Models for Integrated Capabilities. Proceedings of the International Conference on Machine Learning (ICML 2024).

---

> ### Author Response · Authors · 2024-08-28
>
> Dear Reviewer,
>
> Thank you for your effort and thoughtful feedback on our paper. We have carefully addressed your concerns in our rebuttal and hope our responses meet your expectations. As the discussions window is ending, please feel free to raise any additional questions. We value your consideration and respectfully ask you to adjust your scores accordingly if you are content with our responses.

---

### Official Review · Reviewer_4AmE · 2024-07-25
**WildVision-Arena and WildVision-Bench**

**Rating:** 8
**Confidence:** 3
**Correctness:** Good correctness.
**Clarity:** Yes

**Review:**

This paper introduces WildVision-Arena, a rigorous and user-friendly evaluation platform, and WildVision-Bench, a high-quality benchmark for assessing VLM models based on human preferences in real-world interactions. The work is notable for its meticulous design and commitment to open-source data, although further detail on dataset construction and future plans for expansion and maintenance would enhance its impact.

**Strengths:**

- The design of the WildVision-Arena proposed in this paper is rigorous, with a user-friendly interface. The authors have developed a meticulous ranking algorithm tailored for real-world testing scenarios, which better reflects the strengths and weaknesses of VLM models in practical use.
- The authors have carefully constructed WildVision-Bench, which can help community users quickly evaluate their VLM models in real-world scenarios based on human preferences.
- The authors have committed to open-sourcing their data for future VLM research.
- This paper is well-written and easy to follow.

**Additional Feedback:**

No additional comments at this time.

**Documentation:**

Yes

**Ethics:**

No. The authors paid careful attention to the construction of their datasets.

**Limitations:**

The authors adequately addressed the limitations and potential negative societal impact of their work.

**Opportunities For Improvement:**

- Will the authors continue to maintain and support more VLM models in the future, and do they plan to further expand their benchmark to include evaluations in more specialized domains?
- The testing platform, evaluation metrics, and benchmark presented in the paper are rigorous. It would be beneficial to provide more insights into the limitations of existing VLM models and offer more in-depth analysis.
- Will the authors consider expanding their dataset further, and will they release and update new datasets and test sets? Additionally, what type of license will the authors consider using for future releases?
- Can the authors provide a clearer discussion on the collection, partitioning, and construction of the dataset?

**Relation To Prior Work:**

Yes, there is clearly discussed.

**Summary And Contributions:**

This paper primarily benchmarks existing VLMs based on human preferences in real-world multimodal interactions. The authors have developed an open evaluation platform to compare the real user preferences of top-performing VLMs. They analyzed over 20K real-world interactions, providing new insights into the current VLMs. Additionally, they constructed a high-quality benchmark, named WildVision-Bench, which aligns well with human preferences, facilitating rapid assessment of the alignment level of existing VLMs with human preferences.

---

> ### Author Response · Authors · 2024-08-18
> **Response to Reviewer 4AmE**
>
> Thank you very much for your thoughtful feedback, and for recognizing the importance of our proposed VLMs benchmark in real-world applications, the usefulness of our developed open evaluation platform, and the insights from our analysis of 20K interactions. We will address your concerns as follows:
>
> **1. Maintenance of WildVision-Arena**
>
> Since the submission of this paper, the user base of WildVision-Arena has continued to grow, and we have been steadily adding more user-friendly features. We now have 28 VLMs on the leaderboard, which is 6 more models than we had at the time of submission. We will continue to track newly released VLMs and incorporate them into our arena.
>
> In the paper, we presented evaluations under fine-grained image domains and question categories. We will display this information on our platform to better illustrate the VLMs' capabilities in specialized domains. We will also add more subdomains as additional real-world data becomes available on our arena platform.
>
> **2. Further Analysis of Current VLMs' Limitations**
>
> In our released chat and voting data, we provide a more comprehensive list of failure cases and real-world interactions that highlight these models' limitations, in addition to the examples shown in the paper. We are also adding a new tab to showcase these failure cases and maintaining a blog to provide more insightful analysis.
>
> **3. Expanding the Dataset and Its License**
>
> Yes, our dataset will be expanded, updated, and released regularly, with careful ethical considerations.
>
> Regarding the license for data release, we intend to use the Creative Commons Attribution (CC BY) license, which we believe will ensure accessibility to our work.
>
> **4. Collection, Partitioning, and Construction of the Dataset**
>
> At the collection stage, we perform a safety check on our data to avoid collecting any NSFW content.
> At the partitioning stage, we select the hard cases for our benchmark test split.
> At the construction stage, we filter out similar images and text prompts and ensure image and text safety using NSFW tools, followed by a manual check by our experts.

---

### Official Review · Reviewer_f9Sh · 2024-07-27
**Interesting evaluation suite for large vision-language models**

**Rating:** 8
**Confidence:** 3
**Correctness:** As far as I can see, the claims made …
**Clarity:** The paper is written well.

**Review:**

The quality of the paper, as well as clarity and originality, allow for sufficient significance for the work in the respective track. As human feedback on real tasks is involved, the significance is fostered further.

**Strengths:**

The data set covers a wide range of question categories and image domains. The evaluation environment allows easy human evaluation and, with its playful aspects, enhances the chances of collecting larger amounts of human feedback. The results of different models also show important limitations.

**Additional Feedback:**

None.

**Documentation:**

The details about data collection and curation were given appropriately.

**Ethics:**

Potential bias is discussed in the appendix, data sources are named.

**Limitations:**

Unfortunately, the authors discuss the limitations (especially bias in the data) and societal impact only in the work's appendix. Further information in the appendix would be very good to see in the paper already, but probably due to side count restrictions, that is not the case.

**Opportunities For Improvement:**

The related work section could be presented earlier to better understand the extension this paper is providing.

**Relation To Prior Work:**

Related work should be given earlier, so it's easier to understand the contributions.

**Summary And Contributions:**

The authors propose a data set and an evaluation suite for large vision-language models. Human evaluators are invited to compare models against each other in different tasks, yielding a competitive environment and running with evaluation measures known from sports.

---

> ### Author Response · Authors · 2024-08-18
> **Response to Reviewer f9Sh**
>
> We sincerely appreciate your thoughtful review and are grateful for your recognition of the novelty, significance, and overall quality of our proposed dataset and evaluation suite. Your positive feedback reinforces our belief in the value of this work, and we are committed to addressing your insightful suggestions to further enhance the paper.
>
> **1. Incorporating Limitations and Societal Impact into the Main Paper**
>
> We acknowledge your concern regarding the placement of the discussion on limitations and societal impact. Due to page constraints, we initially relegated these discussions to the appendix. However, we agree that these topics are critical to the overall understanding of our work. In response to your suggestion, we will integrate a more comprehensive discussion of these aspects into Section 4 of the main paper. This will ensure that readers can fully grasp the potential implications and challenges associated with our dataset and evaluation suite. Should the paper be accepted, the additional content page will provide us with the opportunity to expand on these discussions, thereby enhancing the paper's depth and clarity.
> Once again, we appreciate your valuable feedback and are encouraged by your positive evaluation of our work. Your suggestions will undoubtedly help us to refine our paper and ensure that it makes a meaningful contribution to the field.

---

### Decision · Program_Chairs · 2024-09-26

**Decision:**

Accept (Poster)

**Comment:**

The majority of reviewers support the paper, recognizing the dataset's timeliness and importance. While one reviewer rejected the paper, there were no major critical issues raised, and the reviewer did not engage further with the authors. Given the overall positive feedback and support from most reviewers, I recommend accepting the paper.